# A ductile chromium–molybdenum alloy resistant to high-temperature oxidation

Frauke Hinrichs[1], Georg Winkens[1], Lena Katharina Kramer[1], Gabriely Falcão[1], Ewa M. Hahn[1], Daniel Schliephake[1], Michael Konrad Eusterholz[1,2], Sandipan Sen[1], Mathias Christian Galetz[3], Haruyuki Inui[4,5], Alexander Kauffmann[6✉] & Martin Heilmaier[1]

Even with the rapid development of renewable energy sources, improving the efficiency of energy conversion from fossil or synthetic fuels remains a challenge because, for example, combustion engines in long-range aircraft will still be needed in the upcoming decades[1]. Increasing their operating temperatures (1,050–1,150 °C (refs. 2–4)) is one option. This requires replacing single-crystalline Ni-based superalloys in the hottest sections of turbines by refractory-element-based materials, which exhibit much higher solidus temperatures beyond 2,000 °C (refs. 5–7). Here we introduce a single-phase Cr-36.1Mo-3Si (at.%) alloy that meets, for the first time, to our knowledge, the most important critical requirements for refractory-element-based materials: (1) relevant resistance against pesting, nitridation and scale spallation at elevated temperatures, minimum up to 1,100 °C, and (2) sufficient compression ductility at room temperature. Although strength and creep resistance in such alloys were already superior to Ni-based superalloys in several cases, oxidation/corrosion resistance, mandatory to withstand the combustion atmosphere, and ductility/ toughness, needed for damage tolerance and device setting, still pose barriers for the development or application of refractory-element-based candidate materials. Any previous successful attempts to address the otherwise catastrophic oxidation of Mo and nitridation of Cr during oxidation suffered from a loss in ductility at ambient temperatures.

The replacement of state-of-the-art Ni-based superalloys in high-temperature applications to improve the efficiency of energy-conversion systems by new metallic–intermetallic materials is at present obstructed by two main limitations: (1) a lack of oxidation resistance and/or (2) a lack of ductility at room temperature (RT). No accurate predictive simulation capabilities for either of the two properties exist at present and, thus, the community still relies on disruptive observations.

For example, Mo and most of its alloys suffer from catastrophic oxidation ('pesting') above 500 °C owing to oxidation to $MoO_3$, which then evaporates as a result of its high vapour pressure[8]. By contrast, Cr is usually considered a passivating element that forms dense $Cr_2O_3$ scales. However, Cr-based alloys suffer from scale spallation and nitridation (ingress of N) when exposed to air at high temperatures beyond approximately 1,000 °C (refs. 9,10). Any previous successful attempt to address oxidation resistance in these alloy systems came with a substantial deterioration in ductility[11,12]. These attempts included the development of high Al-containing and Cr-containing complex concentrated, refractory-element-based alloys with single-phase microstructure that exhibit extreme oxidation resistance, even at temperatures up to 1,500 °C (ref. 13). However, the addition of Al promotes crystallographic ordering with a complete loss of ductility at ambient temperature.

Furthermore, alloys with high amounts of intermetallic phases containing passivating elements were introduced. These intermetallic phases, such as silicides, are brittle in nature. Particularly, Cr–Mo–Si two-phase, silicide alloys[12,14] were identified to be resistant against pesting and nitridation. The protective effect of a $Cr_2O_3$ layer formed in these cases depends decisively on the formation of Si oxide at the interface between the outer $Cr_2O_3$ scale and the metallic substrate. Notably, the potentially ductile, disordered solid solution of the alloy in ref. 12 also exhibited a continuous $Cr_2O_3$ scale.

Hence, a single-phase (Cr,Mo,Si) solid solution with Cr-36.1Mo-3Si (at.%) composition was synthesized by arc melting featuring Cr/Mo = 1.7 ratio similar to the alloy reported in ref. 12. 3 at.% Si is intended to be in solid solution to avoid silicide formation. The choice is consistent with experimentally verified Si contents in solid solution from refs. 12,15. Also, a Si-free version Cr-37.2Mo (at.%) of the same Cr/Mo ratio was synthesized to reveal the role of Si in scale formation and passivation.

## Constitution and microstructure

In Fig. 1a,b, scanning electron microscopy backscattered electron (SEM-BSE) micrographs of the dendritic as-cast microstructures

[1]Institute for Applied Materials (IAM-WK), Karlsruhe Institute of Technology (KIT), Karlsruhe, Germany. [2]Karlsruhe Nano Micro Facility (KNMFi), Karlsruhe Institute of Technology (KIT), Eggenstein-Leopoldshafen, Germany. [3]High Temperature Materials, DECHEMA-Forschungsinstitut, Frankfurt am Main, Germany. [4]Department of Materials Science and Engineering, Kyoto University, Kyoto, Japan. [5]Center for Elements Strategy Initiative for Structural Materials (ESISM), Kyoto University, Kyoto, Japan. [6]Institute for Materials (IM), Ruhr University Bochum (RUB), Bochum, Germany. ✉e-mail: alexander.kauffmann@rub.de

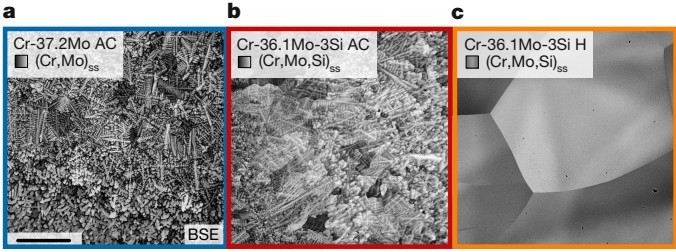

**Fig. 1 | Microstructures of the investigated alloys. a,b,** Dendritic, single-phase microstructure after casting (AC) of Cr-37.2Mo (**a**) and Cr-36.1Mo-3Si (**b**). **c,** Homogenized microstructure of Cr-36.1Mo-3Si H. Scale bar, 500 μm.

(labelled AC) are depicted for Cr-37.2Mo and Cr-36.1Mo-3Si, respectively. A continuous transition in atomic number contrast between the dendrite cores and the interdendritic regions is noted. Chemical compositions were confirmed and deviations from the average owing to the dendritic microstructures are listed in Table 1. The minimum Cr/Mo ratio obtained is 1.30 to 1.51 in the dendrite centres. The maximum Cr/Mo ratio in the interdendritic regions is 2.05 to 2.13. Si exhibits strong enrichment in the interdendritic regions of the Si-containing solid solution. The dendrite spacings are ($30 \pm 10$) and ($52 \pm 14$) μm for Cr-37.2Mo and Cr-36.1Mo-3Si, respectively. Interstitial contamination of N and O are maximum of 450 and 750 at.-ppm (95 and 170 wt.-ppm), respectively. The contamination levels are low and representative of the arc-melting method. The microstructure of Cr-36.1Mo-3Si is homogenized (labelled H) by a heat treatment at 1,600 °C (corresponding to 0.91 $T_S$; $T_S$ is the solidus temperature) for 48 h (Fig. 1c). Although the dendrites are completely removed, substantial grain coarsening to a grain size beyond 500 μm occurred. The crystal structures are, in all cases, identified by X-ray diffraction (XRD) as single-phase, body-centred cubic (bcc) W prototype, Strukturbericht designation A2 (Extended Data Fig. 1). The cubic lattice parameters and crystallographic densities are approximately $a = 3.00$ Å and $\rho = 8.3$ g cm$^{-3}$, in agreement with previously reported lattice parameters and densities[16]. All analyses indicate single-phase microstructures and no secondary phases, specifically, no silicides (Extended Data Fig. 2), in any of the alloys and processing conditions.

## Macroscopic properties

### Oxidation

Because oxidation and pesting resistance are key requirements for new high-temperature refractory metal alloys, the materials were tested in cyclic oxidation. It represents harsher conditions over isothermal oxidation, as superimposed thermal cycling will reveal potential adherence problems of the scales. To assess potential pesting, testing was performed at 800 °C for up to 100 h. Oxidation experiments at 1,100 °C were conducted to investigate the possible application limit for chromia-forming alloys. In Fig. 2a, the area-specific mass change is plotted against time. The mean values and uncertainties from at least three samples per composition are depicted, unless stated otherwise.

For the Si-free Cr-37.2Mo, continuous specific mass loss is recorded from the first hour of oxidation at 800 °C. After 100 h, the specific mass change of the samples is ($-8.1 \pm 1.8$) mg cm$^{-2}$. This specific mass change is much smaller than for Mo-based alloys suffering from pesting, which would result in specific mass change of more than $-100$ mg cm$^{-2}$ and complete disintegration of the material[17]. Instead, the Si-free Cr-37.2Mo AC stays intact and some loose, greenish scale after prolonged oxidation up to 100 h at 800 °C is found. The macroscopic sample geometry changes only marginally during the tests. However, the specific mass change indicates that the Si-free alloy does not form a protective scale at 800 °C and evaporation of volatile $MoO_3$ still occurs. At 1,100 °C, Cr-37.2Mo exhibits high mass loss within a short

time, accompanied by the disintegration of the sample. Oxidation at 1,100 °C is, thus, beyond the capability limit of the Si-free Cr-37.2Mo.

For Cr-36.1Mo-3Si, reproducible specific mass changes are recorded at both 800 and 1,100 °C, irrespective of the microstructural condition AC and H. At 800 °C, the specific mass changes are barely detected, with ($+0.07 \pm 0.04$) and ($+0.04 \pm 0.08$) mg cm$^{-2}$ for AC and H, respectively. The small mass change hints to an absence of substantial evaporation at 800 °C. After 100 h, a dark oxide scale formed. At 1,100 °C, Cr-36.1Mo-3Si AC shows a slightly discontinuous specific mass change. This is indicative of a complex interplay between mass loss owing to evaporation and mass gain owing to scale growth. After 100 h, the specific mass change is $-3$ to $-4$ mg cm$^{-2}$. Cr-36.1Mo-3Si H exhibits a slightly different trend, with a specific mass gain of ($+1.1 \pm 0.4$) mg cm$^{-2}$ at 100 h. Dark grey oxide layers and several spots of green oxides are found. Irrespective of the microstructural condition, the sample geometry remains unchanged and no disintegration is observed for oxidation up to 100 h at both 800 and 1,100 °C.

Si addition is thus proved to exhibit an outstanding positive effect on the specific mass change at 800 °C. Furthermore, it considerably improves the oxidation behaviour at 1,100 °C, as has been shown for binary Cr solid solutions with Si (ref. 9). The Si effect is present regardless of the microstructural condition, dendritic AC or homogenized H. High volume fractions of silicide phases as commonly established in other Mo-based and Cr-based alloys[5,9,18,19] are, hence, not required for oxidation resistance.

## Mechanical properties

Representative true stress–true strain ($\sigma_t - \varepsilon_t$) data are shown in Fig. 2b for RT and 900 °C in compression. These tests were deliberately stopped for microstructural analysis. Selected tests on the oxidation-resistant Cr-36.1Mo-3Si AC and H conditions were performed beyond the maximum stress at RT in compression (Extended Data Fig. 3). The plastic strains detected at maximum stress are 9–15% and 4–6% in AC and H, respectively. Although some cracking occurred during the tests, none of the specimens disintegrated, even when >15% plastic strain was applied.

Deformation occurs serrated immediately from the onset of plastic deformation for all test temperatures. Up to 800 °C, the stress decrease magnitude is between 20 and 50 MPa and is much lower at 900 °C, only 10 MPa.

Owing to the discontinuous plastic flow, the offset strength at 1% plastic strain $\sigma_{1\%}$ was chosen for robust evaluation of strength[20] and is presented in Fig. 2c as a function of test temperature. A large $\sigma_{1\%}$ at RT of approximately 1,100 MPa is observed for the AC conditions. The strength of both alloys in the AC condition decreases until a plateau of approximately 900 MPa is reached at 400 °C (0.30–0.33 $T_S$) and is maintained up to 700 °C (0.44–0.47 $T_S$). Strength substantially decreases beyond 700 °C, at which creep-controlled deformation sets in. However, at 900 °C (0.53–0.57 $T_S$), still 760 MPa, equivalent to 70% of the RT strength, is noted. The detected strength values are identical within the uncertainty ranges for both AC alloys, indicating no notable impact of Si. By contrast, the offset yield strength of Cr-36.1Mo-3Si H is much lower at RT (665 MPa). Grain-boundary strengthening estimated from the Hall–Petch constant of pure Cr (ref. 21) is less than 100 MPa and cannot account for the observed difference in plateau strength to AC. At 900 °C, strength is still high (590 MPa) and strongly decreasing beyond 900 °C. Uncommon to other disordered bcc solid solutions[22], a pronounced strength plateau is obtained for the H condition from RT up to 700 °C.

Owing to the competition of geometrical softening of the sample (cross-sectional reduction) and intrinsic work hardening of the material, tensile ductility can be limited by the localization of plastic deformation if no other sources of failure occur. In the early stages of plastic deformation, work hardening with $\frac{d\sigma_t}{d\varepsilon_t} > \sigma_t$ compensates for the cross-sectional reduction in tension. Plastic deformation occurs

**Table 1 | Compilation of the chemical and physical properties (alloy compositions, contamination levels, lattice parameters, densities, liquidus and solidus temperatures, as well as solidification intervals)**

| | | Cr-37.2Mo | | | Cr-36.1Mo-3Si | | | |
|---|---|---|---|---|---|---|---|---|
| | | AC | | | AC | | | H |
| | | Total | Dendrite core | Interdendritic | Total | Dendrite core | Interdendritic | Total |
| SEM-EDS | Cr (at.%) | 62.1 | 56.5 | 67.7 | 60.7 | 59.1 | 64.8 | 63.0 |
| | Mo (at.%) | 37.9 | 43.5 | 32.3 | 36.6 | 39.0 | 31.6 | 34.0 |
| | Si (at.%) | – | – | – | 2.7 | 1.9 | 3.6 | 3.0 |
| | rel. Cr/Mo | 1.6 | 1.3 | 2.1 | 1.6 | 1.5 | 2.0 | 1.8 |
| HCGE | O (at.-ppm) | 670±340 | – | – | 770±170 | – | – | 243±42 |
| | N (at.-ppm) | 460±20 | – | – | 440±10 | – | – | <144 |
| XRD | $a$ (Å) | 3.003 | – | – | 2.998 | – | – | 3.000 |
| | $\rho$ (g cm$^{-3}$) | 8.38 | – | – | 8.27 | – | – | 8.26 |
| Simulation | $T_L$ (°C) | 2,061 | – | – | 2,007 | – | – | 1,999 |
| | $T_S$ (°C) | 1,941 | – | – | 1,784 | – | – | 1,782 |
| | $\Delta T$ (K) | 120 | – | – | 223 | – | – | 217 |

HCGE, hot carrier gas extraction.

homogeneously under this condition. Beyond $\frac{d\sigma_t}{d\varepsilon_t} = \sigma_t$ (ref. 23), the tensile test becomes unstable and localization of plastic deformation (necking) limits ductility. To evaluate a potential limitation of tensile ductility of the alloys by small work-hardening capability, the true work hardening $d\sigma_t/d\varepsilon_t$ is compared with $\frac{d\sigma_t}{d\varepsilon_t} = \sigma_t$ in Fig. 2d. Besides the aforementioned 9–15% plastic strain in compression up to maximum stress, a high work-hardening capability with $\frac{d\sigma_t}{d\varepsilon_t} > \sigma_t$ is obtained up to 6% plastic strain as the basis for tensile ductility for Cr-36.1Mo-3Si AC. However, 4–6% in the H condition of the same alloy at maximum stress in compression and less work hardening might not be sufficient to obtain relevant tensile ductility.

Unlike most silicide-containing alloys, both solid solutions exhibit compressive ductility at RT in combination with a high strength level up to 900 °C. High work hardening is revealed here as one of the fundamental prerequisites of tensile ductility.

## Microscopic mechanisms
### Oxidation mechanisms

When oxidized, corundum prototype (Strukturbericht designation D5$_1$; Extended Data Fig. 4) Cr$_2$O$_3$ was verified by XRD and scanning electron microscopy energy-dispersive X-ray spectroscopy (SEM-EDS) for all scales examined in this study. For the Si-containing Cr-36.1Mo-3Si, a continuous, well-adherent oxide layer of uniform thickness (Fig. 3a) was obtained for AC and H oxidized at 800 °C for 100 h. The scale thicknesses are (1.7 ± 0.3) and (3.9 ± 2.4) μm (Extended Data Fig. 6d), respectively. This is in strong contrast to the porous, layered microstructure shown in Extended Data Fig. 5 with a total thickness of (91 ± 24) μm formed on the Si-free Cr-37.2Mo under the same conditions. On the basis of the thickness, the area-specific mass gains by the Cr$_2$O$_3$ growth $\frac{\Delta m^{scale}}{A}$ (Extended Data Fig. 6d) can be estimated. The scale growth accounts for specific mass gain of (+0.9 ± 0.2) or (+2.0 ± 1.2) mg cm$^{-2}$ in Cr-36.1Mo-3Si AC and H, respectively, and is thus similar to the experimentally obtained $\frac{\Delta m^{exp}}{A}$ of (+0.01 ± 0.03) and (+0.00 ± 0.03) mg cm$^{-2}$. A slight mass loss by MoO$_3$ evaporation in the transient stage of oxidation is unavoidable; however, it does not affect the long-term behaviour of the alloy. By contrast, the substantial difference between the expected $\frac{\Delta m^{scale}}{A} = (+47 \pm 12)$ mg cm$^{-2}$ and the negative experimental $\frac{\Delta m^{exp}}{A} = (-5.6 \pm 0.4)$ mg cm$^{-2}$ indicate considerable MoO$_3$ evaporation in the case of Cr-37.2Mo. As well as the mass-change observations, the analysis of scale thicknesses at intermediate times allows for an assessment of the growth kinetics (Extended Data Fig. 6). Consistent with the MoO$_3$ evaporation and the porous, non-protective oxide scale on the Si-free Cr-37.2Mo, a linear scale

growth behaviour is obtained. In accordance with the pronounced reduction of the evaporation, the Si-containing Cr-36.1Mo-3Si exhibits a parabolic Cr$_2$O$_3$ growth, with $(15 \pm 10) \times 10^{-14}$ and $(5.9 \pm 1.6) \times 10^{-14}$ cm$^2$ s$^{-1}$ for AC and H, respectively. This proves a diffusion-controlled growth of a passivating scale on Cr-36.1Mo-3Si at 800 °C.

The formation of Cr$_2$O$_3$ is associated with substantial Cr outward diffusion causing a Mo enrichment of the subscale region. This is detected by SEM-EDS and also by XRD. The oxide scales formed on Cr-36.1Mo-3Si after 100 h at 800 °C are thin enough to examine the subscale substrate regions. Two markedly different peaks can be detected (Extended Data Fig. 4). The lattice parameters of 3.125 and 3.138 Å correspond to Mo contents of about (90 ± 2) and (96 ± 2) at.% in AC and H, respectively[16]. As Mo is insensitive to N uptake, the Mo enrichment below the oxide scale acts as a barrier against nitridation, as confirmed for other Cr–Mo–Si alloys[12,24].

In ref. 12, a thin Si oxide layer on solid solution regions of the two-phase alloy had been identified by transmission electron microscopy. As SEM-EDS did not reveal sufficient evidence of Si oxide formation in Fig. 3a, targeted atom probe tomography was performed (Fig. 3c). The upper part of the reconstruction is within the oxide layer and composed of Cr$_2$O$_3$. Towards the substrate and consistent with the other investigations, isolated Mo-enriched solid solution particles are found. A non-negligible O content was detected in these particles; however, no oxidation of Mo to MoO$_3$ takes place. Apart from this, SiO$_2$ is verified within Cr$_2$O$_3$; see the quantifications in Fig. 3c.

In summary, the relevant features identified for oxidation resistance at 800 °C according to Fig. 3a are: (1) the thin, continuous Cr$_2$O$_3$; (2) the Mo-enriched subsurface layer; and (3) the formation of SiO$_2$ at the interface.

A beneficial influence of Mo on the oxidation resistance of Cr alloys has been observed previously in entirely intermetallic Cr-30Mo-30Si consisting of Cr$_3$Si and Cr$_5$Si$_3$ (ref. 24) and in the Mo-lean, two-phase Cr-7Si-2Mo consisting of a (Cr,Mo,Si) solid solution matrix and (Cr,Mo)$_3$Si (ref. 14). From the results in ref. 24, it was concluded that Mo addition to Cr–Si alloys promotes the formation of SiO$_2$, consistent with the present study. For Cr-7Si-2Mo, specific mass gains after 100 h at 1,200 °C are slightly reduced to +5.5 mg cm$^{-2}$ compared with +8 mg cm$^{-2}$ for a Mo-free reference alloy with the composition Cr-9Si (ref. 14). A solid solution layer with an increased Mo content is observed underneath the Cr$_2$O$_3$ top layer. The Mo-enriched region is free of nitridation, whereas Cr$_2$N is detected below the Mo-enriched zones[14]. This is strictly not the case for Cr-36.1Mo-3Si in the present investigation.

The SEM-EDS maps of the scales on Cr-36.1Mo-3Si after oxidation at 1,100 °C for 100 h are shown in Fig. 3b. Continuous white lines mark

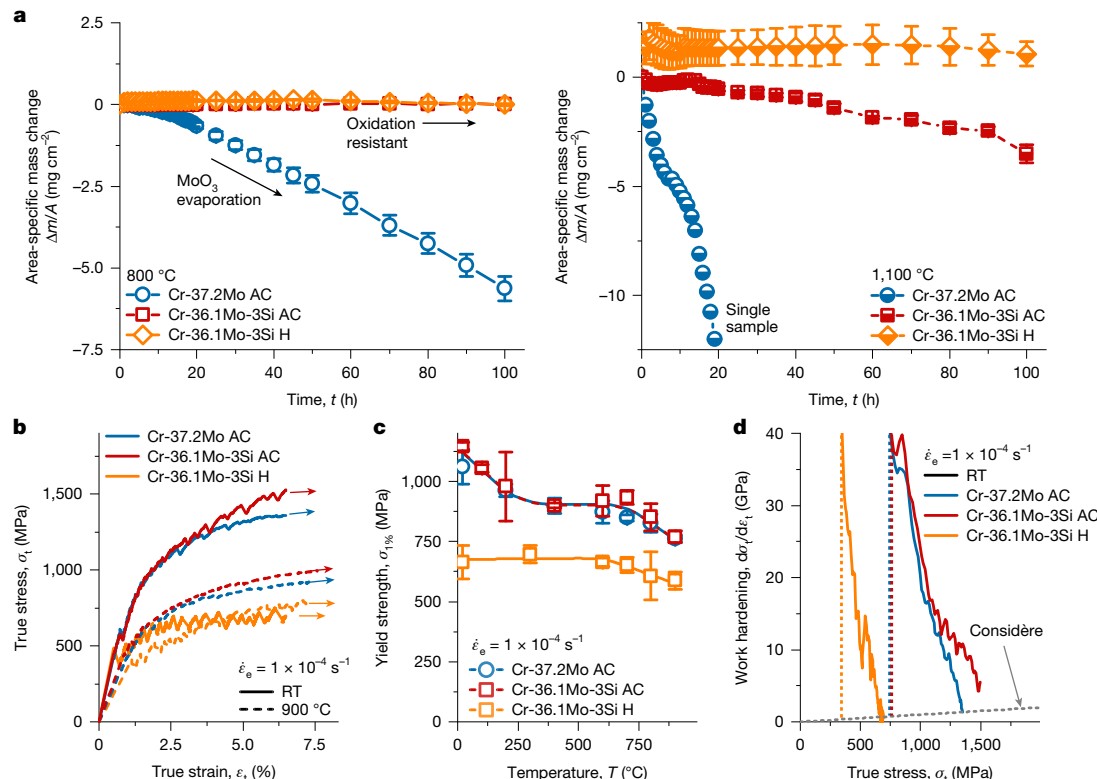

**Fig. 2 | Macroscopic assessment of material properties. a**, Area-specific mass change during cyclic oxidation at 800 °C (left) and 1,100 °C (right). **b**, Representative true stress–true strain curves obtained from compression tests conducted at RT and 900 °C. **c**, One per cent offset yield strength as a function of temperature. **d**, True work hardening as a function of true stress. At least two samples per test temperature were tested and uncertainty was estimated using the standard deviations.

the outer $Cr_2O_3$, with a total thickness of $(21 \pm 6)$ and $(14 \pm 4)$ μm in AC and H, respectively. Scale adhesion at the interface between the substrate and the oxide scale is obtained, without buckling or cracks being observed. However, the oxidation leads to a complex layered outer $Cr_2O_3$ scale. The scale growth is approximately parabolic, with $(9.3 \pm 3.5) \times 10^{-12}$ or $(6.3 \pm 1.4) \times 10^{-12}$ cm² s⁻¹. The rate constants are similar to the reported value of $(1.1 \pm 0.3) \times 10^{-12}$ cm² s⁻¹ on two-phase Cr-32.2Mo-13.5Si (ref. 12). However, the estimated specific mass change by scale growth $\frac{\Delta m^{scale}}{A}$ after 100 h would be $(+11 \pm 3)$ or $(+7.3 \pm 3.6)$ mg cm⁻² compared with the experimental $\frac{\Delta m^{exp}}{A}$ of $(-3.5 \pm 0.4)$ and $(1.1 \pm 0.6)$ mg cm⁻² for AC and H, respectively. Thus, the scale is not protective and slow but strong evaporation occurs.

Consistent with ref. 15, the outer $Cr_2O_3$ seems to be free of $SiO_2$ and originates from Cr-diffusion-controlled outward scale growth. Closer to the substrate, $SiO_2$ particles are frequently obtained within $Cr_2O_3$, indicating O-diffusion-controlled inward growth of the scale. Below the scale, the Mo-enriched region is marked in the elemental maps (arrow). Internal corrosion is observed up to 60 μm deep into the substrate with Si-enriched and O-enriched particles, probably $SiO_2$. For both microstructural conditions, intragranular Si oxide is found, although its proportion is smaller in AC owing to the finer grain size. The observation of $SiO_2$ formation agrees with results obtained for the oxidation of Cr–Si alloys at 1,200 and 1,300 °C (refs. 9,15,25). Assuming that internal oxidation is only caused by Si oxidizing to $SiO_2$, the partial pressure $p_{O_2}$ can be estimated from the Si and $SiO_2$ equilibrium to $p_{O_2} \approx 10^{-24}$ Pa (Fig. 3d). The Si enrichment at grain boundaries deeper in the substrate Cr-36.1Mo-3Si AC is because of the formation of $(Cr,Mo)_3Si$ particles as Si solubility decreases to low temperatures[15]. Thermal cycling from RT to 1,100 °C enables the formation of $(Cr,Mo)_3Si$ under the presence of nucleation sites such as grain boundaries.

The essential features of the schematic scale structure in Fig. 3b are: (1) the layered $Cr_2O_3$ scale; (2) the Mo-enriched layer just below

the scale; (3) the Si oxide layer at the interface, as well as particles in subsurface regions.

Figure 3d depicts the Ellingham diagram for the relevant oxides, $MoO_3$, $MoO_2$, $Cr_2O_3$ and $SiO_2$. The formation of protective $Cr_2O_3$ competes with the formation of non-protective $MoO_2$ and $MoO_3$. This competition is determined by the $p_{O_2}$, which is established at the scale surface and decreases towards the substrate. The obvious protective effect of the Si addition is the further formation of $SiO_2$ closer to or even within the substrate. The presence of $SiO_2$ accordingly keeps $p_{O_2}$ low enough to promote a preferred oxidation of Cr rather than of Mo.

The scale mock-up obtained after 100 h of cyclic oxidation at 800 °C seems simple enough to quantitatively assess the effect of Si on the preferential oxidation of Cr. The oxidation of Cr atoms becomes accessible by the scale thickness $d^{scale}$ and the specific mass gain associated with the scale growth $\frac{\Delta m^{scale}}{A} \approx \frac{\Delta m^{scale}_{O \, in \, Cr_2O_3}}{A} = \frac{3M_O}{M_{Cr_2O_3}} \rho_{Cr_2O_3} d^{scale}$. The difference between $\frac{\Delta m^{scale}}{A}$ to the experimentally obtained $\frac{\Delta m^{exp}}{A}$ allows for the assessment of mass loss by evaporating species $\frac{\Delta m^{vap}}{A}$, for example, $MoO_3$ with $\frac{\Delta m^{vap}}{A} \approx \frac{\Delta m^{vap}_{Mo \, in \, MoO_3}}{A}$ (ref. 26). In this treatment, contributions by the oxidation of Si to $SiO_2$, porosity of the scale as well as other evaporating species such as $CrO_3$ (not expected for <900 °C (ref. 27)) are neglected:

$$\frac{\Delta m^{exp}}{A} \approx \frac{\Delta m^{scale}_{O \, in \, Cr_2O_3}}{A} - \frac{\Delta m^{vap}_{Mo \, in \, MoO_3}}{A} \quad (1)$$

The ratio of oxidized Cr to Mo ions is then:

$$\frac{n^{oxCr}}{n^{oxMo}} = \frac{2M_{Mo} \rho_{Cr_2O_3} d^{scale}}{3M_O \rho_{Cr_2O_3} d^{scale} - M_{Cr_2O_3} \frac{\Delta m^{exp}}{A}} \quad (2)$$

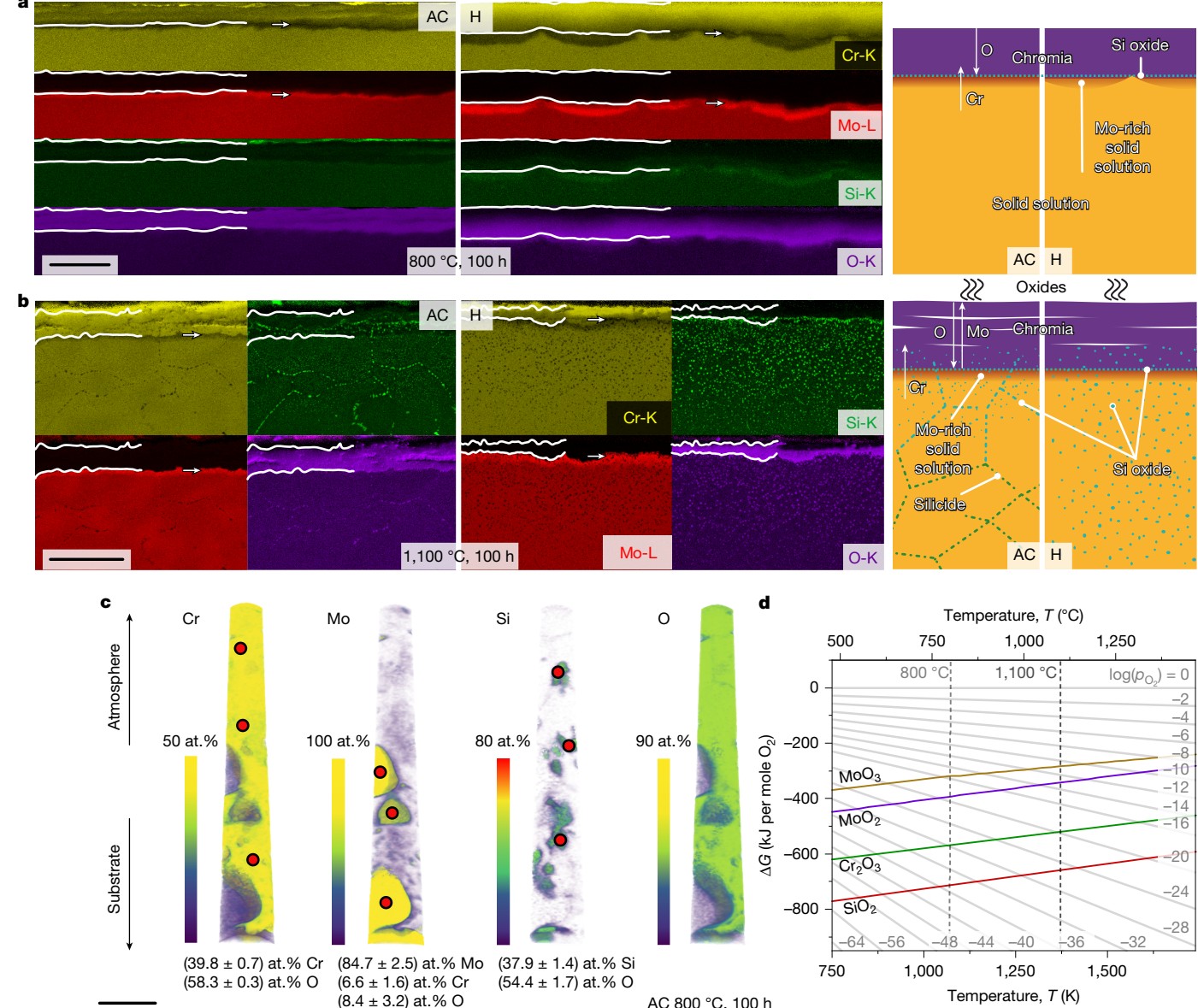

**Fig. 3 | Microscopic appearance of the oxide scales from Cr-36.1Mo-3Si after 100 h of cyclic oxidation. a**, SEM-EDS (left) and schematic (right) scale structure for oxidation at 800 °C. **b**, SEM-EDS (left) and schematic (right) scale structure for oxidation at 1,100 °C. **c**, Atom probe tomography from the substrate–scale interface for oxidation at 800 °C (only AC condition). **d**, Ellingham diagram with Gibbs free energy of formation ($\Delta G$) per mole $O_2$ as a function of temperature. Scale bars, 5 μm (**a**); 50 μm (**b**); 50 nm (**c**).

For the oxidation-resistant Cr-36.1Mo-3Si tested at 800 °C, the assessment yields $\frac{n^{oxCr}}{n^{oxMo}}$ of (5.4 ± 1.1) and (4.3 ± 0.6) for AC and H, respectively. This is substantially higher than the Cr to Mo ratio of 1.7 provided by the alloy. This indicates the preferential oxidation of Cr over Mo. By contrast, only (2.6 ± 0.3) is obtained for the Si-free Cr-37.2Mo. The preference for oxidation of metal ions is, thus, strongly shifted towards Cr by the presence of only 3 at.% Si.

## Deformation mechanisms

Figure 4a–c shows orientation imaging microscopy maps on longitudinal sections of Cr-36.1Mo-3Si deformed at RT and 900 °C up to plastic strains of approximately 6% in compression. Colour-coded maps indicate crystallographic orientation with respect to the compression direction (CD), greyscale maps yield the diffraction pattern quality. Quantitative assessments of the crystallographic orientations in Fig. 4d,e reveal slightly preferred ⟨100⟩/⟨111⟩ directions parallel to the CD. This is caused by the slip plane normal rotation towards the CD by the dislocation slip in the ⟨111⟩ direction on {1$\bar{2}$1} to {1$\bar{3}$2} planes[28].

Besides dislocation-mediated plasticity, deformation twinning is observed as an alternative deformation mechanism. In most grains, several twin systems are activated. Without exception, all identified twin systems are of {112}⟨11$\bar{1}$⟩ type (Extended Data Fig. 7) and correspond to rotations of $(60^{+0.0}_{-2.9})°$ about ⟨111⟩ as the common twin system in bcc metals and alloys[29]. The number density of twins is much lower at 900 °C compared with at RT. Furthermore, the twin lath thickness is larger at 900 °C. Deformation twinning is active regardless of the microstructural condition of Cr-36.1Mo-3Si, that is, AC or H.

The observation of deformation twins coincides with the serrations occurring in compression testing. In contrast to continuous deformation through dislocation slip, entire laths are discontinuously formed during each twinning event, leading to stress drops. However, microcracks are also observed predominantly oriented parallel to the CD. These cracks originate from a limited compatibility of large grains to

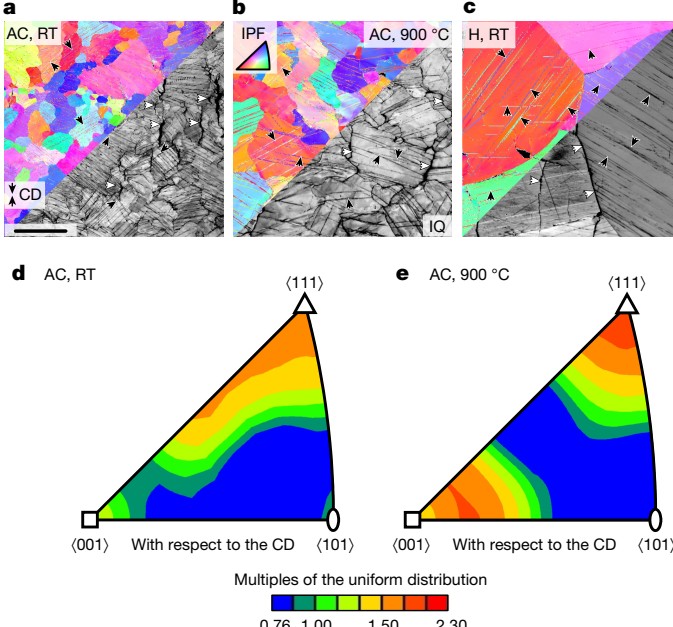

**Fig. 4 | Microstructural changes induced by mechanical testing of Cr-36.1Mo-3Si AC and H in compression up to a plastic strain of approximately 6%. a**, AC tested at RT. **b**, AC tested at 900 °C. **c**, H tested at RT. The colour code in the orientation maps corresponds to the inverse pole figure (IPF) of the CD. The greyscale image parts highlight diffraction pattern quality (IQ). Black arrows indicate deformation twins and white arrows indicate longitudinal cracks along grain boundaries. **d**,**e**, Orientation distribution function presented in the IPF of the CD in multiples of the uniform distribution. Scale bar, 200 μm (**a**).

transversal thickening as well as from stress concentration at terminations of twin laths[30,31]. The role of grain boundaries for premature crack formation in tensile tests needs to be clarified in future studies.

Deformation twinning usually complements dislocation slip as a process for plastic deformation. It is furthermore considered to be a low-temperature and/or high-strain-rate process. Stresses required for twinning are usually larger than for dislocation slip. Thus, the latter is the more commonly observed elementary process of initiation of plastic deformation in bcc metals and alloys. However, there are frequent reports also for deformation twinning in bcc refractory metals and alloys, for example, Mo (refs. 32,33), Cr (ref. 34), V (ref. 35), W (ref. 36), Nb (refs. 37,38), Ta (ref. 39), Cr-35Fe (ref. 34), Cr–Re (refs. 34,40), Mo-35Re (ref. 41) and Nb–V (refs. 20,42), in polycrystalline[33–35,38–41,43] or single-crystalline[32,36,37,41] conditions. As the two fundamental deformation mechanisms complement each other depending on the internal stress level, a complex interdependence of intrinsic and extrinsic parameters determines their activation, for example: (1) solid solution strengthening; (2) grain size; (3) temperature (and strain rate); and (4) work hardening.

(1) Deformation twinning is often reported in bcc solid solutions rather than pure metals, potentially because of increasing onset stress for dislocation slip by solid solution strengthening. Reid and Gilbert[34] reported no twinning for Cr-4.9Re and Cr-15.2Re but did for Cr-35Re. Similarly, ref. 41 reported deformation twinning in Mo-35Re but not in Mo. Owing to the large difference in lattice parameter and the large shear moduli for Mo and Cr, a beneficial influence of solid solution strengthening on the activation of deformation twinning is plausible[44].

(2) Available data for polycrystalline Mo (ref. 33), Cr (ref. 34), V (ref. 35) and Cr-15Re (ref. 40) indicate a smaller twinning stress with increasing grain size (the literature also reports counterexamples, in which twinning occurred preferably at an intermediate grain size (Nb (ref. 38)) or where twinning stress increases with increasing grain

size (Cr-35Re (ref. 40))). Thus, the occurrence of twinning for the present samples with a large grain size of >100 μm is more likely than for fine-grained samples. According to ref. 45, the twinning in Cr is more sensitive to grain size as compared with slip, such that the onset stress of slip easily surpasses the twinning stress at large grain sizes. Thus, initiation of plastic deformation might be achieved by twinning as opposed to slip. If we associate the first stress drops here with the twinning stress being reached[36–38], it is approximately equal for both alloys and between 350 and 500 MPa for all temperatures. However, local stress concentration, for example, at grain boundaries or twin tips, might still be sufficiently high to initiate local dislocation slip. Thus, twinning might be the predominant mechanism of early plastic deformation but is probably not the only one. Conversely, if the twinning stress is not too large as in the present case, local stress concentrations or work hardening can also lead to twinning in an alloy that predominantly deforms by slip.

(3) The temperature dependence of yield strength depicted in Fig. 2c resembles the dependencies on (1) solid solution strengthening and (2) grain size. In the coarse-grained Cr-36.1Mo-3Si H, the very large grain size favours twinning to be the predominant deformation mechanism at the onset of plastic deformation. As the onset stress of twinning is a process insensitive to temperature and strain rate[46], the temperature dependence of yield strength up to 700 °C remains weak. By contrast, the AC alloys exhibit smaller grain sizes obstructing deformation twinning. Consequently, dislocation slip exhibits a higher proportion to plastic deformation with twinning still being active. Hence, the strong temperature dependence of yield strength typical of bcc metals and alloys[22] dominated by thermal activation of dislocation kink pair formation is obtained (AC in Fig. 2c).

As twinning is considered a low-temperature deformation mechanism, the twinning propensity obtained at 900 °C (Fig. 4b) seems unusual at first glance. However, Weaver[43] suspected twinning in Cr and Cr-1V at temperatures of 200–600 °C but did not report microstructural or crystallographic proof. Deformation twinning at elevated temperatures was verified for up to 600 °C in Nb-55V (ref. 20) and 1,000 °C in MoWRe (ref. 47). The alloys investigated here surpass the binary Nb–V solid solutions and show deformation twinning at higher homologous temperature compared with MoWRe.

(4) As deformation twins are high-angle grain boundaries, they impede dislocation motion. Deformation twinning thus might contribute to dynamic grain-boundary strengthening similar to the observations during twinning-induced plasticity[48]. This might rationalize the high work hardening shown in Fig. 2d, specifically in the fine-grained AC conditions. On the basis of the discussion above, the proportion of slip is larger for the fine-grained material compared with the coarse-grained H. Hence, the work-hardening capability by the dynamic refinement and impediment of slip is improved in a fine-grained condition.

## Consequences for materials development

The present candidate materials development relies on the use of refractory metal elements that exhibit characteristic problems in oxidation, limiting their application. The choice of Cr and Mo addresses these problems: Cr leads to protective $Cr_2O_3$ scale formation, whereas the Mo enrichment in the subsurface zone converts these regions to be resistant against nitridation. Si as a minor third element guarantees the slow growth of the $Cr_2O_3$ scale by gettering O and avoiding oxidation of Mo to $MoO_3$. An optimal ratio of Cr, Mo and Si and the impact of further minor alloying elements need to be identified to examine the temperature limits of these characteristics.

The low amount of Si allows for the synthesis of single-phase, disordered solid solutions. Apart from this fundamental prerequisite to obtain ductility, the appearance of both dislocation slip and deformation twinning is confirmed, the latter contributing to a very good work-hardening behaviour. The deformation behaviour at very

different grain sizes highlights the microstructure optimization potential by tailoring the competition of twinning and slip.

The materials presented here show the simplest manufacturing conditions with the relevant property profile already being met. All further optimization treatments known from classical metallurgy might also be applied to the Cr−Mo−Si system to achieve improved and application-relevant properties, for example, thermomechanical treatment of cast metallurgical material or powder metallurgically processed material. The results gained in compression in the present study need to be confirmed on microstructural conditions with tailored grain size, to obtain a balance of the contributions by slip and twinning on the one hand. This is specifically relevant to mitigate crack initiation by twinning[30,31]. On the other hand, it is relevant to reveal the role of grain boundaries or segregation for potential limitation of tensile ductility.

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

## Methods

Several batches of the alloys with the desired compositions of Cr-37.2Mo (at.%) and Cr-36.1Mo-3Si were manufactured from elemental Mo 99.95% (Plansee SE), Cr 99+% (EVOCHEM) and Si 99.99% (chemPUR) in a standard arc-melting process using an AM/0.5 arc melter (Edmund Bühler GmbH). Melting was performed in Ar (>99.998%) atmosphere in a water-cooled Cu crucible. The pressure during arc melting was 600 mbar. Before each remelting step, the melting of a piece of Zr was performed to bind residual O. The ingots of 100–150 g were flipped and remelted five times to ensure homogeneity. One batch of Cr-36.1Mo-3Si was also subjected to a homogenization treatment at 1,600 °C for 48 h with 100 K h$^{-1}$ heating and cooling rate and under flowing Ar atmosphere in a HTRH 70-600/18 resistance tube furnace supplied by Carbolite Gero GmbH & Co. KG. The ingots were subsequently cut by electrical discharge machining into cuboidal samples of dimensions $5 \times 5 \times 4$ mm$^3$ for oxidation experiments and $5 \times 3 \times 3$ mm$^3$ for compression testing.

All surfaces of oxidation samples were ground with SiC paper of P2500 grit size. The surface area of the samples was obtained by optical macroscopy (Wild Heerbrugg M420 and Olympus Stream Enterprise 1.7). Cyclic oxidation was performed at 800 and 1,100 °C in laboratory air using muffle furnaces (Carbolite Gero GmbH & Co. KG). The duration of the oxidation cycles started with 1-h cycles for the first 20 h of oxidation. Afterwards, the cycle duration was increased to 5 h until the total oxidation time was 50 h, followed by 10-h cycles until the samples were exposed for 100 h in total. The samples were oxidized in Al$_2$O$_3$ crucibles and weighed in the crucibles between oxidation intervals. Initial mass and mass changes of the oxidation samples were tracked after each oxidation cycle using a precision balance with an accuracy of 1 µg (Sartorius). The mass change per total surface area of the unoxidized sample was calculated from the data of at least three samples per test temperature. Average specific mass changes were determined from these samples and uncertainty was estimated using the standard deviation. Oxide scale thicknesses were determined manually by processing SEM-BSE micrographs of cross-sections using the open-source image-processing software ImageJ (version 1.53c) after 1, 10 and 100 h of cyclic oxidation[49]. The oxide layer was examined with respect to its morphology on the substrate, the types of oxide grown and the element distribution within the oxide layer.

Standard deviations of the experimental quantities $\Delta \frac{\Delta m^{\mathrm{exp}}}{A}$ and $\Delta d^{\mathrm{scale}}$ were obtained from several specimens and from micrographs of four sides of a single specimen, respectively. These quantities were used to estimate the standard deviation of $\frac{n^{\mathrm{oxCr}}}{n^{\mathrm{oxMo}}}$ (see equation (2)) by uncertainty propagation. The assessment yields

$$\Delta \frac{n^{\mathrm{oxCr}}}{n^{\mathrm{oxMo}}} = \sqrt{\left( \frac{\partial}{\partial d^{\mathrm{scale}}} \frac{n^{\mathrm{oxCr}}}{n^{\mathrm{oxMo}}} \Delta d^{\mathrm{scale}} \right)^2 + \left( \frac{\partial}{\partial \frac{\Delta m^{\mathrm{exp}}}{A}} \frac{n^{\mathrm{oxCr}}}{n^{\mathrm{oxMo}}} \Delta \frac{\Delta m^{\mathrm{exp}}}{A} \right)^2}$$

$$= 2 \sqrt{\frac{M_{\mathrm{Cr_2O_3}}^2 M_{\mathrm{Mo}}^2 \rho_{\mathrm{Cr_2O_3}}^2 \left( \left( d^{\mathrm{scale}} \Delta \frac{\Delta m^{\mathrm{exp}}}{A} \right)^2 + \left( \frac{\Delta m^{\mathrm{exp}}}{A} \Delta d^{\mathrm{scale}} \right)^2 \right)}{\left( 3 M_{\mathrm{O}} \rho_{\mathrm{Cr_2O_3}} d^{\mathrm{scale}} - M_{\mathrm{Cr_2O_3}} \frac{\Delta m^{\mathrm{exp}}}{A} \right)^4}} \tag{3}$$

The assessment in the present case (cyclic oxidation at 800 °C for 100 h) is, within the standard deviations, robust against the choice of reference, for example, (1) single or several specimens used for obtaining specific mass change as well as (2) the distinct strategy to obtain the scale thickness on cross-sections, namely the number of reading and sampled surfaces.

The $3 \times 3$-mm$^2$ surfaces of the compression test samples were prepared by grinding with SiC paper of P2500 grit size using a self-constructed sample holder to ensure parallel contact faces. The dimensions of each specimen were measured using a digital indicator with 1 µm resolution (Sylvac SA). Compression tests were performed using a Zwick UPM 1478 universal testing device (ZwickRoell) with an induction coil to heat the samples to testing temperatures of up to 900 °C. Compression tests at 1,100 °C were performed using a Zwick Z100 universal testing device equipped with a vacuum furnace provided by Maytec (Maytec Mess- und Regeltechnik GmbH). To minimize friction between the sample surfaces and punches, boron nitride was applied to the sample surfaces. All tests (100, 200, 400, 600, 700, 800, 900, 1,100 °C) were started after a sufficient holding time of 15 min at each test temperature to ensure that the temperature was evenly distributed. The initial engineering strain rate was set to $1 \times 10^{-4}$ s$^{-1}$. Displacement was measured by a capacitive strain gauge attached to the Al$_2$O$_3$ punches as well as an inductive strain gauge tracking the displacement of the crosshead. The signals from the two strain gauges were compared so that malfunctions or device-specific measurement artefacts could be interpreted as such. Stress–strain data were corrected for the non-linear initial loading range using the maximum slope detected for the elastic deformation. As plastic deformation is serrated/discontinuous in the present case, stress at 1% plastic offset strain was evaluated for all samples. Engineering stress–strain $\sigma_{\mathrm{e}}$–$\varepsilon_{\mathrm{e}}$ data are converted to true stress–true strain $\sigma_{\mathrm{t}}$–$\varepsilon_{\mathrm{t}}$ data assuming volume conservation and prismatic sample shape. At each temperature, a minimum of two samples of each composition were tested.

Samples for microstructural analysis in the as-cast, homogenized, oxidized or deformed conditions were cold-mounted in VariKEM 200 resin (Schmitz-Metallographie GmbH), ground to grit P2500 with SiC paper, followed by 3-µm and 1-µm diamond polishing steps. Surface finish was achieved using a colloidal OP-S suspension (Buehler ITW). SEM-BSE micrographs of the as-cast and the oxide scales were taken at 20 kV acceleration voltage on a LEO Gemini 1530 (Carl Zeiss AG) scanning electron microscope.

SEM-EDS and scanning electron microscopy electron backscatter diffraction (SEM-EBSD) were performed on an Auriga 60 (Carl Zeiss AG) scanning electron microscope equipped with an Octane Super EDS detector and an EDAX DigiView EBSD camera (Ametek). For SEM-EDS analysis, the scanning electron microscope was operated at 16 kV acceleration voltage. Standard-free, ZAF-corrected SEM-EDS maps were created to visualize the element distributions in the oxide layers and close-to-surface substrate region. The K lines for Cr, Si and O and the L line for Mo were used to quantify the element distribution. For SEM-EBSD, the scanning electron microscope was set to 20 kV acceleration voltage and a 120-µm aperture was used. EDAX TEAM was used to acquire the EBSD maps using a W prototype phase for indexing. Overview maps are $840 \times 840$ µm$^2$ in size with a hexagonal grid of 2-µm step size. Detailed maps of twinned regions were acquired with a size of $150 \times 150$ µm$^2$ at a step size of 500 nm. The indexing rate was about 125 s$^{-1}$ with a successful indexing rate above 98%. The twin elements were identified using pole figures of the {112} habit plane with ⟨11$\bar{1}$⟩ shear direction and {1$\bar{1}$0} shear plane normal. The surface traces of the twin boundaries were considered in the analysis as well.

For atom probe tomography investigations of the chromia scale and the scale–substrate interface, tip specimens were prepared by a conventional lift-out method[50] using an Auriga 60 scanning electron microscope equipped with a dual-beam focused ion beam (Zeiss AG). The experiments were performed in a LEAP 4000X HR (Cameca Instruments) in pulsed laser mode. A laser wavelength of 355 nm with a pulse energy of 50 pJ and frequency of 100 kHz was used. The base temperature of the sample was kept at 50 K, with the detection rate set to 0.003 ions per pulse. Data reconstruction and analysis were done using the software package AP Suite 6.3.1 (Cameca Instruments). With respect to the quantification shown in Fig. 3c, the following aspects need to be considered. The presence of particles with different evaporation fields induces local variations in the tip curvature and, thus, local variations in the electric field. The resulting trajectory aberrations may lead to mixing of the particles and the surrounding matrix in the reconstruction[51]. Also, the lateral resolution in LEAP 4000 instruments

is restricted to 1 nm (ref. 52), which results in overlapping positions of particles and matrix atoms in the present case. To reduce these effects, small analysis volumes were selected to only examine the cores of the particle. Another factor to consider for the present analysis is the evaporation of O as neutral molecules[53], which has been reported for different metal oxides and leads to a characteristic underestimation of O in both oxides in the quantification, for example, $Cr_2O_3$ with a O to metal ratio of 1.46 (expected: 1.5) and $SiO_2$ with a ratio of 1.4 (expected: 2.0).

Crystal structures and lattice parameters were determined through XRD. The diffraction patterns were recorded in Bragg–Brentano geometry with $\Theta/\Theta$ focusing using a D2 phaser (Bruker Corp.) equipped with a LYNXEYE line detector. The X-ray tube with Cu-Kα radiation was operated at 30 kV and 10 mA. Scans were performed in the range $2\Theta = 10°–145°$. The step size was 0.01° in $2\Theta$, with an accumulated acquisition time of 384 s per step. The sample was rotated during measurement to improve statistics. For phase identification, diffraction patterns were analysed using the open-source software PowderCell, version 2.4 (ref. 54). The cubic lattice parameter of the W prototype crystal structures of the solid solutions were determined by extrapolating the measured lattice parameters towards $\Theta = 90°$ by using the weight function $\frac{1}{2}(\cot^2\Theta + \cot\Theta \cdot \cos\Theta)$, similar to the approach suggested by Nelson and Riley[55].

The CALPHAD software package Pandat (version 2023) with the proprietary databases PanHEA and PanMo (version 2023) was used to determine the solidus temperature for each alloy investigated here. The results for the two databases are the same. For this, the calculations were performed for thermodynamic equilibrium by means of the PanEngine.

## Data availability

The data presented in this study are available from KITopen at https://doi.org/10.35097/vaagh6pu00hqunu7 (ref. 56) under a CC BY-SA 4.0 license. Further information is available on request to the corresponding author. Source data are provided with this paper.

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

**Acknowledgements** We gratefully acknowledge financial support by the Deutsche Forschungsgemeinschaft (DFG) within the framework of research training group RTG 2561 MatCom-ComMat. We acknowledge the support from the Alexander von Humboldt Foundation for their cooperative research conducted under the Humboldt Fellowship of H.I. Furthermore, H.I. appreciates the support from JSPS KAKENHI (grant nos. JP22H00262 and JP23K17338). This work was partly carried out with the support of the Karlsruhe Nano Micro Facility (KNMFi; www.knmf.kit.edu), a Helmholtz Research Infrastructure at Karlsruhe Institute of Technology (KIT; www.kit.edu). We acknowledge the chemical analysis by HCGE at the Institute for Applied Materials (IAM-AWP) by T. Bergfeldt, Karlsruhe Institute of Technology (KIT). We thank B. Beichert and C. Wehnes for their experimental support.

**Author contributions** F.H.: investigation, formal analysis, data curation, visualization, project administration, writing – original draft, writing – review and editing. G.W.: formal analysis, software, writing – original draft, writing – review and editing. L.K.K.: investigation, formal analysis, data curation, writing – review and editing. D.S.: investigation, formal analysis, data curation, visualization, writing – original draft, writing – review and editing. G.F.: investigation, formal analysis, writing – review and editing. E.M.H.: investigation, formal analysis, writing – review and editing. M.K.E.: investigation, formal analysis, writing – review and editing. S.S.: investigation, formal analysis, writing – review and editing. M.C.G.: supervision, funding acquisition, writing – review and editing. H.I.: funding acquisition, writing – review and editing. A.K.: investigation, methodology, formal analysis, data curation, visualization, supervision, writing – original draft, writing – review and editing. M.H.: supervision, resources, funding acquisition, writing – review and editing.

**Funding** Open access funding provided by Ruhr-Universität Bochum.

**Competing interests** The authors declare no competing interests.

**Additional information**
**Correspondence and requests for materials** should be addressed to Alexander Kauffmann.

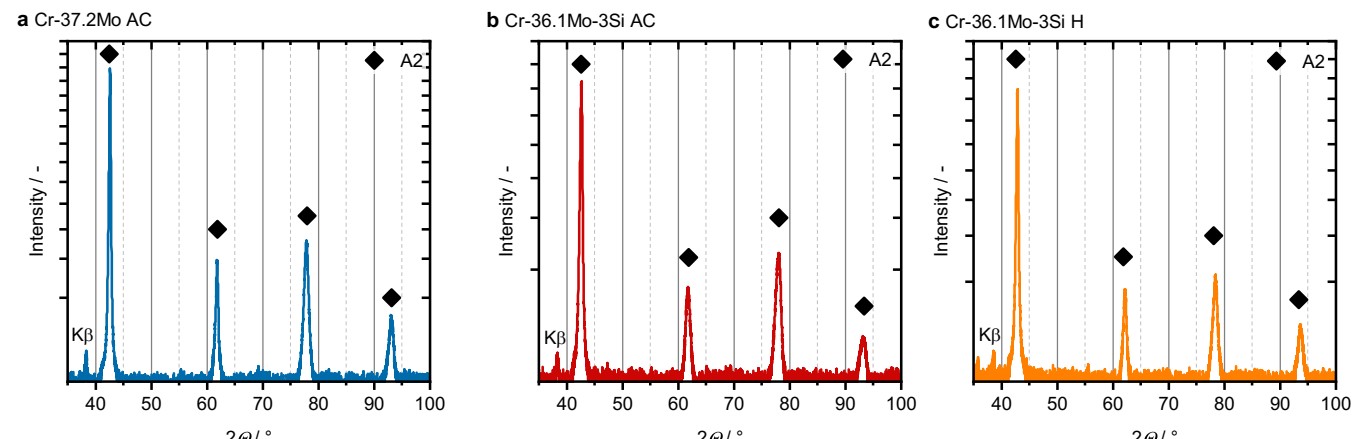

**a** Cr-37.2Mo AC

**b** Cr-36.1Mo-3Si AC

**c** Cr-36.1Mo-3Si H

**Extended Data Fig. 1 | Confirmation of the A2 crystal structure of the conditions investigated by XRD. a**, Cr-37.2Mo AC. **b**, Cr-36.1Mo-3Si AC. **c**, Cr-36.1Mo-3Si H. A2 is the Strukturbericht designation of the disordered bcc W prototype. The materials are single-phase A2. The cubic lattice parameters and crystallographic densities are determined as $a$ = 3.003 Å and $\rho$ = 8.42 g cm$^{-3}$ for Cr-37.2Mo and $a$ = 2.998 Å and $\rho$ = 8.27 g cm$^{-3}$ for Cr-36.1Mo-3Si in the AC condition, respectively. No relevant changes are obtained after homogenization. Compared with previously reported lattice parameters for similar solid solutions[16], no deviations are detected. Si was previously noted to have no marked influence on the lattice parameter of these solid solutions[57,58].

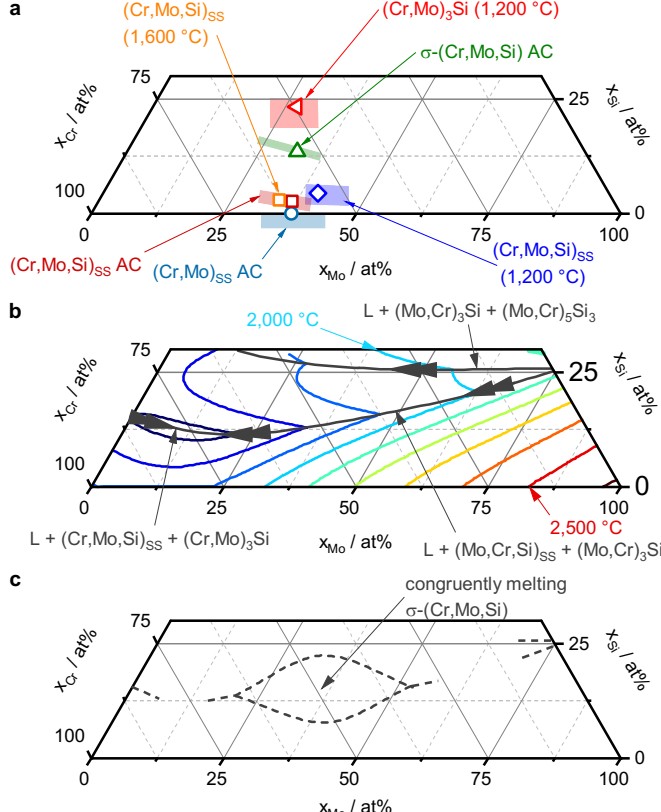

**Extended Data Fig. 2 | Compilation of experimental and thermodynamic data in the vicinity of the investigated alloys. a**, Experimentally verified phases from the present work as well as from refs. 12,16. AC indicates dendritic as-cast condition, with the shaded regions bridging the compositions in the dendritic and interdendritic regions. Phases from homogenized or two-phase conditions are also shown with the respective temperatures of the heat treatment. Shaded regions indicate microstructural variations and measurement uncertainties. **b**, Liquidus projection from Pandat using PanHEA and PanMo. **c**, Suggestion for the correction of the phase diagram/thermodynamic assessment based on experimental data not considered at present on the stable phases. Neither the databases nor the most up-to-date thermodynamic assessment in ref. 59 include the Strukturbericht D8$_b$ σ-(Cr,Mo,Si) phase, first reported by Rudy and Nowotny in 1974 (ref. 16), which served as the basis of the development in ref. 12. The successful synthesis of single-phase σ-(Cr,Mo,Si) phase in Cr-32.2Mo-13.5Si from the liquid proves congruent solidification and the according lenticular solidification region of σ-(Cr,Mo,Si) in the liquidus projection in **c**. The presence of this ternary phase needs to be considered in improved thermodynamic calculations for future guided alloy development.

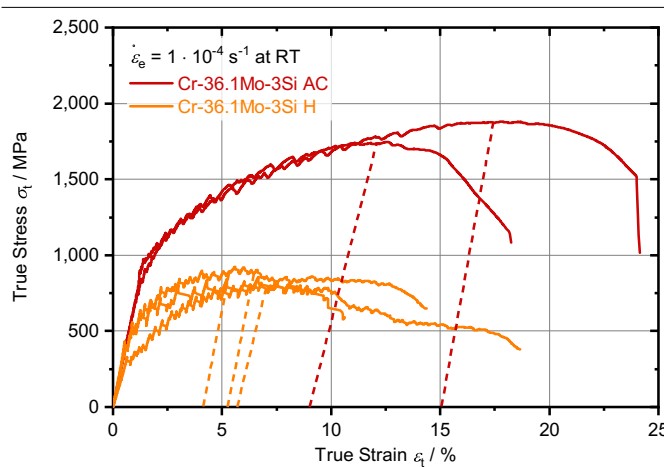

**Extended Data Fig. 3 | Compression test results on Cr-36.1Mo-3Si strained beyond maximum stress.** The samples did not fully disintegrate. Plastic strains detected at maximum stress (dashed lines) are 9% and 15% as well as 4% to 6% in AC and H, respectively. Grain-boundary strengthening estimated from the Hall–Petch constant of pure Cr of 800 MPa$\sqrt{\mu m}$ (ref. 21) cannot account for the difference between the fine-grained AC and coarse-grained H plateau strength. The grain-boundary-strengthening contribution only changes from approximately 80 (AC) to 20 MPa (H).

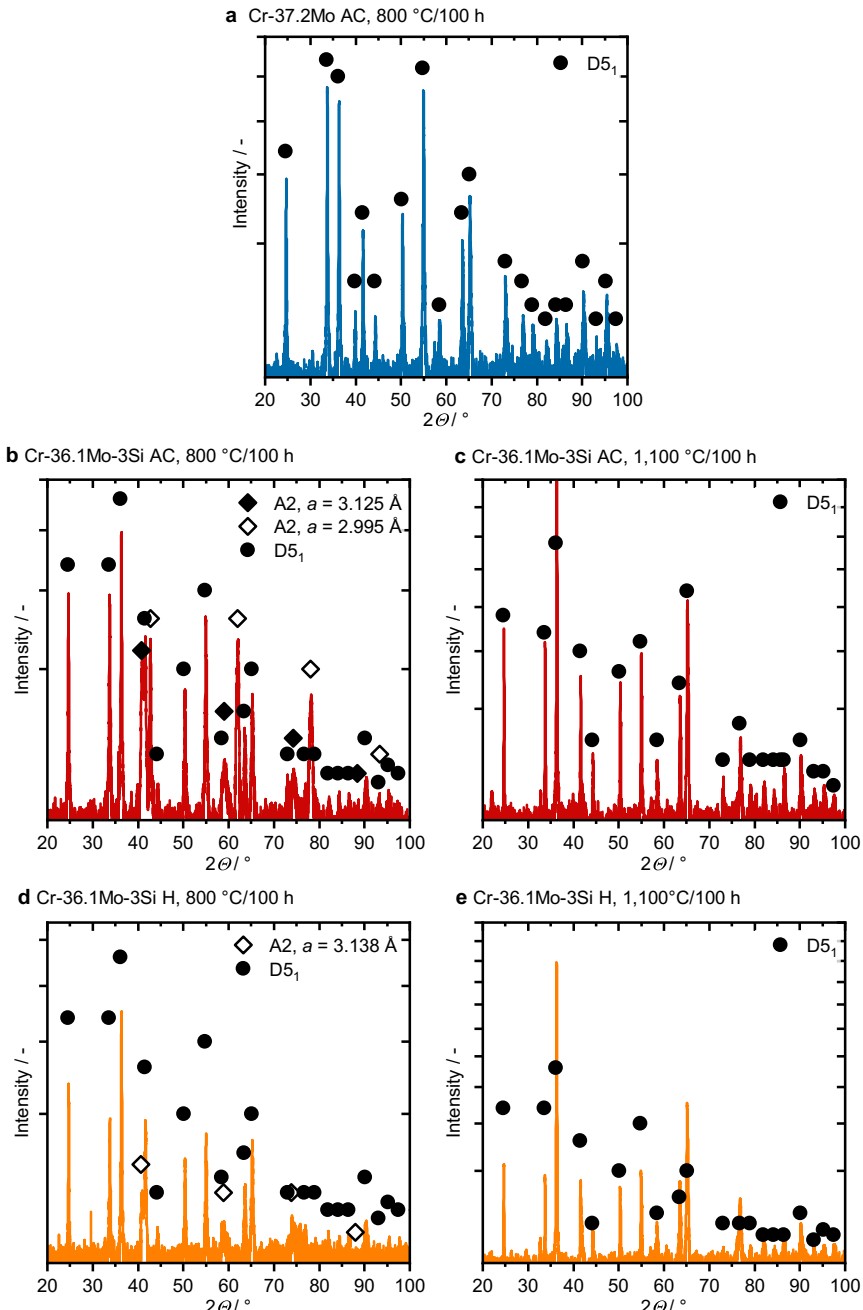

**Extended Data Fig. 4 | Confirmation of the D5₁ crystal structure of the oxide scales formed during 100 h of cyclic oxidation. a**, Cr-37.2Mo AC tested at 800 °C. **b**,**c**, Cr-36.1Mo-3Si AC tested at 800 and 1,100 °C. **d**,**e**, Cr-36.1Mo-3Si H tested at 800 and 1,100 °C. D5₁ is the Strukturbericht designation of the corundum prototype.

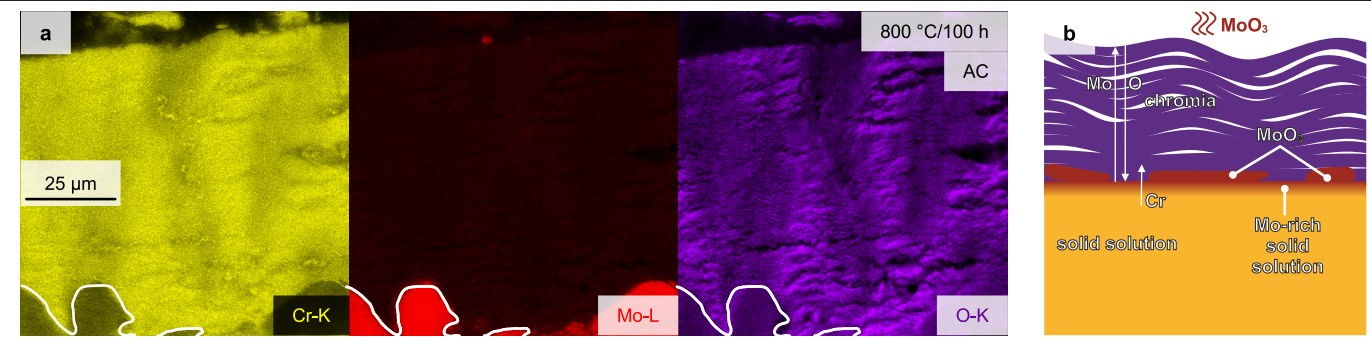

**Extended Data Fig. 5 | Oxide scale and near-surface region of Cr-37.2Mo AC after 100 h of oxidation at 800 °C. a**, SEM-EDS. **b**, Schematic.

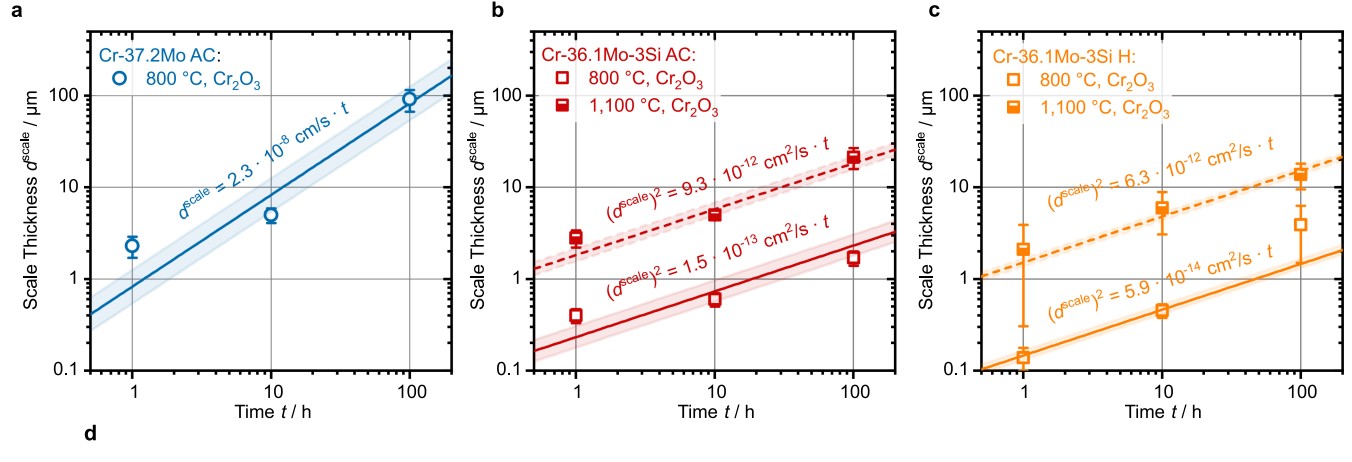

**d**

| Alloy | Condition | $T$ / °C | Type of fit | Growth rate constant $k_{d,n}$ cm$^n$/s | Exponent $n$ | Pearson's $r$ | $R^2$ |
|---|---|---|---|---|---|---|---|
| Cr-37.2Mo | AC | 800 | free | $2.0 \cdot 10^{-7}$ | 1.3 | 0.92 | 0.84 |
| | | | $n = 1$ | $2.3 \cdot 10^{-8}$ | 1.0 | | 0.73 |
| Cr-36.1Mo-3Si | AC | 800 | free | $1.9 \cdot 10^{-18}$ | 3.2 | 0.97 | 0.93 |
| | | | $n = 2$ | $1.5 \cdot 10^{-13}$ | 2.0 | | 0.61 |
| | | | $n = 3$ | $1.1 \cdot 10^{-17}$ | 3.0 | | 0.93 |
| | | | $n = 4$ | $8.1 \cdot 10^{-22}$ | 4.0 | | 0.90 |
| | H | | free | $9.8 \cdot 10^{-13}$ | 1.7 | 0.95 | 0.94 |
| | | | $n = 2$ | $5.9 \cdot 10^{-14}$ | 2.0 | | 0.93 |
| | | | $n = 3$ | $2.2 \cdot 10^{-18}$ | 3.0 | | 0.78 |
| | | | $n = 4$ | $7.9 \cdot 10^{-23}$ | 4.0 | | 0.64 |
| | AC | 1,100 | free | $3.9 \cdot 10^{-13}$ | 2.4 | 0.94 | 0.84 |
| | | | $n = 2$ | $9.3 \cdot 10^{-12}$ | 2.0 | | 0.84 |
| | H | | free | $8.7 \cdot 10^{-14}$ | 2.6 | 0.99 | 0.99 |
| | | | $n = 2$ | $6.3 \cdot 10^{-12}$ | 2.0 | | 0.90 |

**e**

| | Cr-37.2Mo | Cr-36.1Mo-3Si | | | |
|---|---|---|---|---|---|
| | AC | AC | | H | |
| $T$ / °C | 800°C | 800°C | 1,100°C | 800°C | 1,100°C |
| $n$ | 1 (linear) | 2 (parabolic) | | 2 (parabolic) | |
| $k_{d,n}$ / cm$^n$/s | $(2.3 \pm 1.2) \cdot 10^{-8}$ | $(15 \pm 10) \cdot 10^{-14}$ | $(9.3 \pm 3.6) \cdot 10^{-12}$ | $(5.9 \pm 1.6) \cdot 10^{-14}$ | $(6.3 \pm 1.4) \cdot 10^{-12}$ |
| $d^{\text{scale}}$ after 100 h / µm | $91 \pm 24$ | $1.7 \pm 0.3$ | $21 \pm 6$ | $3.9 \pm 2.4$ | $14 \pm 7$ |
| $\frac{\Delta m^{\text{scale}}}{A}$ / mg/cm² | $+47 \pm 12$ | $+0.9 \pm 0.2$ | $+11 \pm 3$ | $+2.0 \pm 1.2$ | $+7.3 \pm 3.6$ |
| $\frac{\Delta m^{\text{exp}}}{A}$ / mg/cm² | $-5.6 \pm 0.4$ | $+0.01 \pm 0.03$ | $-3.5 \pm 0.4$ | $+0.00 \pm 0.03$ | $+1.1 \pm 0.6$ |

**Extended Data Fig. 6 | Oxide scale thicknesses as a function of oxidation time. a**, Cr-37.2Mo AC. **b**, Cr-36.1Mo-3Si AC. **c**, Cr-36.1Mo-3Si H. The lines represent fits to the data. **d**, Summary of fit data of the scale growth according to $(d^{\text{scale}})^n = k_{n,d} \cdot t$, with $n$ being the growth exponent and $k_{n,d}$ the rate constant. Linear fits (instrumental weight was used to account for the individual uncertainty of data points, $R^2$ coefficient of determination is provided,

Pearson's $r$ for the linear correlation is provided) of the linearized form $\log d^{\text{scale}} = \frac{1}{n}(\log k_{n,d} + \log t)$ were performed. **e**, Compilation of cyclic oxidation kinetics data: growth exponent $n$ and rate constants $k_{n,d}$, scale thickness $d^{\text{scale}}$ after 100 h and the estimated $\frac{\Delta m^{\text{scale}}}{A}$ by the scale growth, experimentally determined $\frac{\Delta m^{\text{exp}}}{A}$.

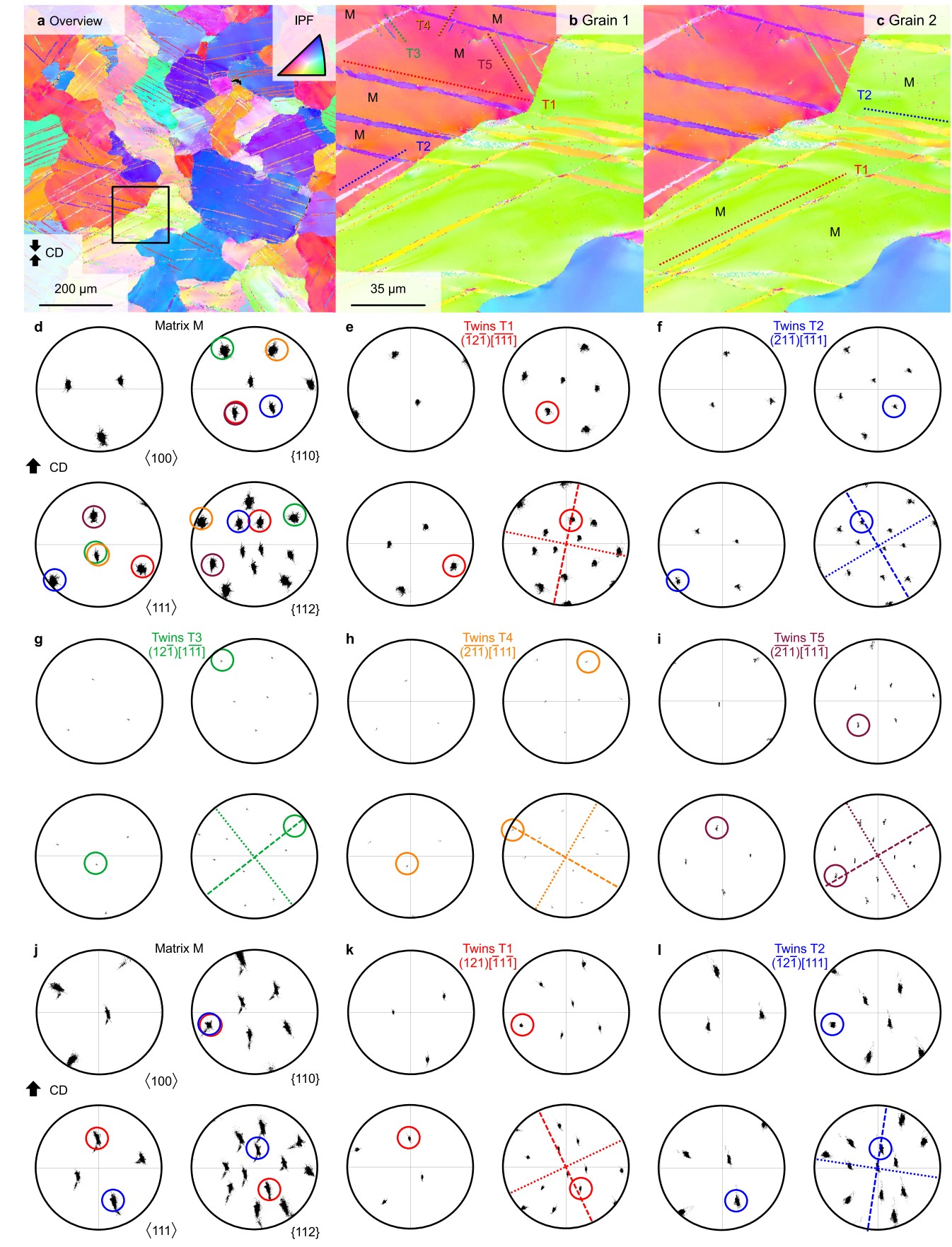

**Extended Data Fig. 7** | See next page for caption.

