## [Peer Review File · Nature]

A ductile chromium-molybdenum alloy resistant to high temperature oxidation

Corresponding Author: Professor Alexander Kauffmann

Version 0:

Reviewer comments:

Referee #1

(Remarks to the Author)

In this manuscript, new results on oxidation behavior and compression properties of Cr-37.2Mo and Cr-36.1Mo-3Si cast alloys (compositions are in at.%) are reported. The microstructure of the alloys in the as-cast condition, after cyclic oxidation at 800°C and 1100°C, as well as after compression deformation was also investigated and reported. The alloys have single-phase BCC crystal structures with almost the same lattice parameters. It was found that the alloy doped with Si exhibits excellent oxidation resistance against pitting, nitridation and scale spallation and also showed improved ductility at room temperature relative to previously reported Cr-Mo-Si alloys containing silicide and/or laves phases. The results are new and important for the advanced alloy development and they should definitely be published. However, I have quite a few questions and comments, which should be addressed. Hope the authors will find them constructive and useful.

- 1) Why the non-equilibrium, as-cast condition was selected for the study? Both the alloys showed large micro-segregation of the alloying elements between dendritic and inter-dendritic regions. Why the alloys were not studied in a homogenized, chemically equilibrium condition? What is the effect of the observed dendritic structure on the studied properties and how the properties change after the homogenization treatment?
- 2) The authors report equilibrium liquidus and solidus temperatures based on CALPHAD modeling. Then they estimated a non-equilibrium solidus temperature based on "the experimentally determined composition of the inter-dendritic regions." (a) It would be worth if the authors describe this estimate method in more details, e.g. in supplementary materials. (b) Why did they not run a non-equilibrium (e.g. Scheil) solidification model? (c) Could the authors report the equilibrium phase diagram for Cr-36.1Mo-3Si to illustrate/confirm that the alloy indeed has a single-phase BCC crystal structure?
- 3) Interpretation of the oxidation behavior is vague. During oxidation at 800°C in both Cr-37.2Mo and Cr-36.1Mo-3Si alloys a Cr₂O₃ scale formed, but in the Si-containing alloy it was much thinner. The author stated that poorer oxidation behavior of Cr-37.2Mo was due to volatility of MoO₃ oxide. The author did not explain, however, what was the reason that MoO₃ did not form in the Si-containing alloy, although the diffusion paths for O and Mo was considerably smaller (thinner Cr₂O₃ layer).
- 4) The oxide growth in Cr-37.2Mo at 1100°C was not examined because of rapid oxidation. However, this study would be very useful for better understanding the effect of Si on the oxidation behavior at 1100°C. Why the oxidation behavior of these two alloys are so much different? I really did not find solid explanation to this in the manuscript.
- 5) Mechanical properties. Why compression deformation stopped after 6% strain? Later the authors state that the samples were tested up to the maximum plastic strain of >6%. What was the actual maximum compression strain? 6% compression strain is too low to claim that the alloys are ductile. It would be great if the authors report at which compression strain (in the range of at least 0-30%) the alloys fail and what was the type of fracture. Generally alloys, which fracture strain during compression deformation is less than 10%, show brittle type of fracture (cleavage or intergranular fracture) and they are brittle during tensile deformation.
- 6) Ductility analysis based on Considere criterion (lines 176-188) can only be applied to ductile materials, which form necking during tensile testing. The authors must prove first that the alloys show ductile type of fracture. For this, please extend deformation until the samples fail and then report compression fracture strain, how the samples fail (explosive, shear or cleaved type) and also provide images of the fracture surfaces. The authors' brief mentioning that micro-cracks were developed parallel to the compression direction (i.e. perpendicular to the tensile stresses) indicates that the alloys are brittle. The authors statement on "exceptional compressive ductility" must be proven by deforming the samples to much larger

strains than 6%.

Other, less critical comments.

7) Line 11 "at low temperatures." Could you be more specific? Do you mean room temperature?

8) Lines 108-120: "The specific mass loss of the samples is between -5.5 and -6 mg/cm²." If you said mass loss, then do not use the negative sign in the respective value. The mass loss of -5.5 actually indicates the mass gain of 5.5 .

9) Lines 189-205 and below. Growth of the Cr₂O₃ scale should result in the mass gain, while the mass loss is observed. Based on the thickness of the scale, one can estimate how much Mo was lost during oxidation in both alloys.

The statement on line 196 "no protective scale has formed" seems to contradict the experiment. A continuous scale of the same Cr₂O₃ forms in both alloy. Should discuss why the scale thickness increases linearly in the binary alloy. Why Cr₂O₃ continuous scale does not slow down oxidation in this alloy. How the MoO₃ evaporates through continuous Cr₂O₃? Why much thinner layer of the same Cr₂O₃ formed in the Si-containing alloy is a more protective scale? Why Mo is less volatile in the Si-containing alloy? All these questions should be discussed/answered.

10) Line 217 "cubic growth law" but line 259 "The scale growth occurs almost parabolic." These statements should be corrected. The authors should provide the oxidation equation used for the analysis: $dn = kt$. Then, in the second instance, the corrected phrase is "The scale growth follows the quadratic law".

11) Lines 231-232: "Mo addition to Cr-Si alloys promotes the formation of SiO₂ consistent with results from the present study." I am not sure if this statement is correct, because SiO₂ was not detected experimentally in the studied alloy at 800°C.

12) Paragraph lines 224-238. The authors did not answer the main question: why MoO₃ does not form or forms in a very small amount in the Si-containing alloy, while this process dominates in the alloy without Si.

13) Lines 284-295. What was the texture of un-deformed samples parallel to the compression axis. I doubt if any noticeable texture would develop after only 6% compression strain.

14) Line 307: "some short microcracks, located predominantly parallel to the compression direction," Crack developed parallel to compression direction (i.e. perpendicular to the tension stresses) after 6% compression strain is an indication of brittleness of the alloy samples.

15) Line 312: "Fundamentally, deformation twinning competes with dislocation slip" These are not competing but complementary processes.

16) Lines 359-366. It is clear that the authors missed the main point: the presence of Si slows down formation of volatile MoO₃. The reason for this should be uncovered and discussed.

17) Materials and Methods section. What was the mass of arc melted samples? Why homogenization treatment was not used to get rid of the dendritic structure?

Referee #2

(Remarks to the Author)

In the paper on "Overcoming barriers for high temperature alloys: oxidation and ductility" the authors present a systematic study of the behavior of Cr-Mo alloys with and without the addition of a minor level of Si. The results of the study would be of interest to the community examining high temperature refractory metal alloys. Before the paper can be considered further there are several issues that require clarification and revision.

First, as the authors point out in the introduction if their alloy is intended to enable an increase in operating temperature beyond the capability of current Ni base superalloys, why did they only conduct the testing up to 1100°C? At 1100°C the Ni base superalloys already perform well. In order to displace the Ni base superalloys, the evaluation must be done at a higher temperature of 1300 or 1400°C. This is an important point. For example, while the chromia scale appears to work in the Cr-36.1Mo-3Si alloy at 1100°C, it is well known that chromia will lose integrity and protection above 1100°C due to the development of volatile CrO₃. In fact, this issue is mentioned on line 281 for 1100°C. Thus, the challenge of overcoming oxidation at high temperature will not be met by the Cr-36.1Mo-3Si alloy.

In figure 1 the as-cast samples show a clear dendritic microstructure. Since there is no indication of a homogenization treatment, it appears that the oxidation testing was done on as-cast samples. However, during the long-time oxidation tests at high temperature the as-cast microstructure can change and the change can confound the analysis of the oxidation behavior. This change can also influence the apparent parabolic and cubic kinetics shown in figure S4 which by the way is not explained in the manuscript. For example, how do the rate constants and exponents relate to the proposed oxidation mechanism in figure 3?

It is not clear if the mechanical tests were conducted in air, argon or vacuum. In figure 4 a and b there appear to be precipitates (silicides?) at some of the grain boundaries which implies according to figure 3d that the sample oxidized. Moreover, while the development of grain boundary cracking may not lead to complete failure under compression, the cracks would reduce the ductility under tension and also result in poor fatigue performance. This behavior does not support the claim of overcoming the ductility challenge.

Unless these points are fully addressed the authors' conclusions can not be accepted.

Version 1:

Reviewer comments:

Referee #1

(Remarks to the Author)

The authors correctly addressed some of my comments in revised manuscript and conducted additional experiments, which is much appreciated. However, the authors responses to my comments 5) and 6) are not satisfactory and I cannot accept

them.

In my comment 5) I pointed out that 6% compression deformation is too low to claim that the alloy is ductile and I suggested the authors to compression deform the alloy sample to fracture and report the fracture strain. This should be a simple experiment, which would provide an important information on the compression ductility and brittleness of the alloy. This information is critical, as the authors claim that the studied Cr-36Mo-3Si alloy is both oxidation resistant and ductile. The second statement should be proven experimentally. Currently, the authors failed to prove this and, therefore, their main authors "achievement," synergy of good oxidation resistance and good ductility, is still questionable.

My comment 6) has not been addressed either. Considere criterion can only be applied to tensile testing of ductile materials and it estimates strain at which plastic instability (necking) may start to form during tensile test. It cannot be applied to the compression tests of a brittle alloy. As I mentioned in my comment, the authors must prove that the alloy shows ductile behavior and ductile type fracture. But they failed to do these simple additional experiments. This additional information is needed for this manuscript to prove the authors statements about "overcoming barriers for high temperature alloys: oxidation and ductility". Otherwise, the paper title is misleading.

I am also not satisfied with the authors responses to my comment 2). From one side, they included the non-equilibrium Scheil assessment, which is much appreciated. On the other side, discrepancy of the Scheil model with experimental results for Cr-Mo-Si they tried to explain by the absence of the thermodynamic data for the Sigma-(Cr,Mo,Si) phase in the used database. However, if the studied Cr-Mo-Si alloy is truly single-phase BCC structure, as the authors state, the absence of some intermetallic phases in the database should not affect the CALPHAD calculations towards the BCC phase.

When replying to the reviewers' comments, it would be nice to also indicate lines in the manuscript where the respective revisions were made, instead of just stating "We clarified the statement" or "We corrected the statement" or "This is addressed in the novel version of the manuscript."

Additional comments related to the revised manuscript:

Lines 11-13. The authors continue to state that the Cr-36Mo-3Si shows "sufficient ductility at low temperatures" and thus shows "outstanding and yet unmet property." First, please be specific and replace "low" with "room". Second, 6% compression strain, which also includes elastic strain, is not sufficient to state that the alloy shows "sufficient ductility" at room temperature. Will the alloy show yielding during tensile testing or it will fracture during elastic loading? I bet the latter will happen. At what compression strain it fractures? What are the fracture modes for this alloy? I asked similar questions in my first review. The authors provide no answer and no additional experiments to disprove that the alloy is actually brittle.

Lines 51-54: I would agree that the oxidation behavior of the Cr-Mo-Si is outstanding; however, I disagree with the statement that the alloy shows "sufficient ductility at low temperatures." The authors did not prove that the alloy shows "outstanding and yet unmet property combination" (oxidation and ductility).

Lines 95-103. This is incorrect interpretation of the discrepancy between CALPHAD calculations and the experiment. If BCC is an equilibrium phase, then absence of any other phases from the database should not affect the result: Gibbs free energy of the BCC phase should not increase relative to the Gibbs free energy of the other phases, present in the database due to the absence of some of the other phases!

Table 1. Please provide the explanation why the amount of O and N decreased after annealing in Ar at 1600C/48 h.

Lines 135, 137 (related to my comment 8) in the first review). If you write "loss" then the value must be positive. Otherwise, the loss of -6 mg/cm² indicates +6mg/cm² mass increase!

Sentence in Lines 156-157 should be revised. If spallation and evaporation events are absent, then what is responsible for the "small mass change" of this alloy at 800C? How was the mass change measured? Was the mass of spalled material included or it was measured separately, using the mass of the remaining sample? This should be described in details in Methods.

Lines 186-191. The authors ignore my comments 5) and 6) in my previous report regarding the validity of the compression tests with low strain levels (not exceeding 6%) and using Considere criterion. I still insist that compression strain should be extended to the fracture event to confirm that the material is indeed ductile. Or please conduct tensile tests and show that the alloys will not fracture during elastic loading.

Line 195: "For all test temperatures up to 800C ..." What is this? The samples were tested at two temperatures, RT and 900C only!

Lines 208-212: What is the reason of twice lower strength at room temperature after homogenization treatment? This noticeable drop in strength cannot be explained by grain growth.

Lines 219-221. This conclusion should be proven using tensile tests. As mentioned above, Considere criterion cannot be applied to compression tests.

Figure 2. Please plot Figure 2(b) in coordinates true stress versus true strain.

Lines 232-234: "the scale is identified as monolithic Cr₂O₃" but "The oxide scale is composed up of porous, layered microstructure of Cr₂O₃." These statements contradict each other. Should be revised. The word "monolithic" is used in several instances in the manuscript.

Lines 360-373. Discussion on the fiber texture is misleading. Answering to my comment 13), the authors said that the samples were extracted from the arc-melted button in random orientations and their texture was not analyzed. Therefore, the texture observed in the compressed sample can be an original texture, not formed after deformation. I would recommend removing all the "texture" data from the manuscript in this case.

Lines 463-473. Oxidation mechanism and role of Si. At 3 at.% Si, could the authors estimate how much oxygen Si can get and prove that this amount of SiO₂ formed can explain the observed reduction of the Cr₂O₃ growth rate and no formation of MoO₃? The current vague explanation of the oxidation process in Cr-36Mo-3Si, as well as insufficient mechanical testing experiments, does not deserve the publication in Nature.

Referee #2

(Remarks to the Author)

In the revised manuscript the authors have addressed all of my review comments. The additional experiments on homogenized samples including the APT measurements have clarified the oxidation behavior. In addition, a note has been added concerning the need to address the performance above 1100°C. This will be a challenge and must include resistance to water vapor attack. The authors have also added a clarification concerning the difficulty in relating compression test behavior to tensile behavior which is another challenge. I appreciate their opinion regarding the replacement of Ni base superalloys by their alloy, but I maintain my opinion that the current performance of their alloy does not qualify as a replacement. Much more development is required for their alloy.

Version 2:

Reviewer comments:

Referee #1

(Remarks to the Author)

The authors addressed all my comments. Although they still did not follow my suggestion to test the samples to larger compression strains, achieve sample failure and determine the modes of fracture (ductile or brittle), which I feel would considerably improve the reader trust to the reported data, I think the manuscript improved considerably in all other aspects.

Nature
Referees

Institute for Applied Materials (IAM-WK)

Head of Business Unit:
Prof. Dr.-Ing. Martin Heilmaier
Prof. Dr. rer. nat. Astrid Pundt
Prof. Dr.-Ing. habil. Volker Schulze

Engelbert-Arnold-Straße 4
76131 Karlsruhe, Germany

Phone: +49 721 608 46594
Email: alexander.kauffmann@kit.edu
Web: www.iam.kit.edu/wk

Official in charge: Alexander Kauffmann
Date: 10/21/2024

Revision of an article

Dear referees,

please find in this electronic re-submission our revised manuscript on “Overcoming barriers for high temperature alloys: oxidation and ductility” (F. Hinrichs et al.).

We greatly appreciate the thorough review by the referees. We conducted the additional research as suggested by the reviewers. We were able to confirm the previously reported results and claims for a novel batch of homogenized material. The mechanical characterization of the additional microstructural condition allowed to further conclude on the different contributions leading to the unexpected deformation behavior of the Cr-Mo-Si alloy. Furthermore, we conducted atomic scale characterization of the scale/substrate interface via atom probe tomography. We were able to finally confirm the formation of SiO₂ as the relevant difference to promote the outstanding oxidation resistance of the Si-containing alloy over the Si-free alloy.

We are convinced that the key results of our study are verified to an extent that allows the community to proceed with the engineering development of this novel type alloy. We have added a forecast to the research and developments that are required to take profit of the observations from our investigations.

Please find the point-by-point description of our revisions and response in the following. The according changes in the manuscript are marked up by red color.

Referee #1 (Remarks to the Author):

In this manuscript, new results on oxidation behavior and compression properties of Cr-37.2Mo and Cr-36.1Mo-3Si cast alloys (compositions are in at.%) are reported. The microstructure of the alloys in the as-cast

condition, after cyclic oxidation at 800°C and 1100°C, as well as after compression deformation was also investigated and reported. The alloys have single-phase BCC crystal structures with almost the same lattice parameters. It was found that the alloy doped with Si exhibits excellent oxidation resistance against pitting, nitridation and scale spallation and also showed improved ductility at room temperature relative to previously reported Cr-Mo-Si alloys containing silicide and/or laves phases. The results are new and important for the advanced alloy development and they should definitely be published. However, I have quite a few questions and comments, which should be addressed. Hope the authors will find them constructive and useful.

The remarks by both reviewers led to additional experiments that further strengthened the interpretation of the results with regard to the underlying mechanisms. *We were able to confirm all relevant statements. The additional results were homogeneously embedded into manuscript.*

1) Why the non-equilibrium, as-cast condition was selected for the study? Both the alloys showed large micro-segregation of the alloying elements between dendritic and inter-dendritic regions. Why the alloys were not studied in a homogenized, chemically equilibrium condition? What is the effect of the observed dendritic structure on the studied properties and how the properties change after the homogenization treatment?

The initial focus of the study was to provide the simplest initial microstructural condition as possible. Even with this microstructural inhomogeneity, the novel Cr-Mo-Si alloy performed exceptional. Specifically with respect to the inhomogeneous Mo distribution (which would rather lead to a worse oxidation behavior based on the expectations on the inferior behavior of all known Mo-based alloys), the performance of the alloy was revealed to be robust under the testing conditions.

From a fundamental viewpoint, we agree that a homogeneous microstructure presents the best subject of investigation. Accordingly, we have extended the study by the homogenized condition of Cr-36.1Mo-3Si. *We were able to confirm the key findings of the initial version of the manuscript. The much larger grain size after homogenization, allowed the further interpretation of the contributions to make deformation twinning a dominant deformation mode (over dislocation mediated plasticity) in the present case.*

We appreciate the remark by the referee and *we are convinced that the extended study provides the relevant starting point for the community to apply all possible microstructural engineering techniques to tailor the Cr-Mo-Si to its maximum performance. We added an assessment on these aspects.*

2) The authors report equilibrium liquidus and solidus temperatures based on CALPHAD modeling. Then they estimated a non-equilibrium solidus temperature based on “the experimentally determined composition of the inter-dendritic regions.” (a) It would be worth if the authors describe this estimate method in more details, e.g. in supplementary materials. (b) Why did they not run a non-equilibrium (e.g. Scheil) solidification model? (c) Could the authors report the equilibrium phase diagram for Cr-36.1Mo-3Si to illustrate/confirm that the alloy indeed has a single-phase BCC crystal structure?

We included the non-equilibrium Scheil assessment of the two alloy compositions to the manuscript. *With respect to the Si-free Cr-Mo alloy, the experimental findings are consistent with equilibrium and non-equilibrium assessment.*

The non-equilibrium assessment of the Si-containing Cr-Mo-Si alloy fails due to the inconsistency of the currently available thermodynamic databases.

Neither the Pandat databases used in our group nor the most recent assessment of the Cr-Mo-Si in literature consider the presence of the Strukturbericht D8_b σ -(Cr,Mo,Si) phase that is close to our alloy compositions. The according invariant reactions in the vicinity of our alloy compositions are therefore wrongly modelled.

We have added a respective description and highlighted the additional effort needed by the Calphad community. (It is relevant to note that this is not new and was already subject of our previous investigations on σ -(Cr,Mo,Si) phase.)

3) Interpretation of the oxidation behavior is vague. During oxidation at 800°C in both Cr-37.2Mo and Cr-36.1Mo-3Si alloys a Cr₂O₃ scale formed, but in the Si-containing alloy it was much thinner. The author stated that poorer oxidation behavior of Cr-37.2Mo was due to volatility of MoO₃ oxide. The author did not explain, however, what was the reason that MoO₃ did not form in the Si-containing alloy, although the diffusion paths for O and Mo was considerably smaller (thinner Cr₂O₃ layer).

We acknowledge the remark by the reviewer. We conducted additional target investigations by atom probe tomography (APT) to investigate the scale/substrate interface. Indeed, we were able to identify the formation of SiO₂, not only at 1100°C as presented in the initial version of the manuscript but also for 800°C oxidation.

The O partial pressures at the scale/substrate interface are therefore confirmed to be so low that oxidation Mo to MoO₃ is not possible under the presence of Si. This is supported by the combined crystallographic and chemical identification of the Mo-rich solid solution which contains some amounts of O; however, by far not enough to allow for the oxidation of Mo to MoO₃.

We have added these results and adopted the manuscript to highlight the confirmation of our initial claim.

4) The oxide growth in Cr-37.2Mo at 1100°C was not examined because of rapid oxidation. However, this study would be very useful for better understanding the effect of Si on the oxidation behavior at 1100°C. Why the oxidation behavior of these two alloys are so much different? I really did not find solid explanation to this in the manuscript.

The Si-free Cr-37.2Mo fails catastrophically within a few hours and minutes at 800 and 1100°C, respectively. A relevant comparison of the remaining residue (there is no comparable scale/substrate interface left even after very short exposure times, the substrate gets consumed by the reaction) does not seem reasonable – the conditions of the subjects of investigation are much different.

We instead focused on the identification of Si distribution after oxidation at 800°C where it was not revealed as O getter till the revision of the manuscript and the additional APT experiments. At 1100°C, the oxidation of Si to SiO₂ was already confirmed in the initial version of the manuscript.

The additional investigations, confirmation and conclusions were added to the manuscript.

5) Mechanical properties. Why compression deformation stopped after 6% strain? Later the authors state that the samples were tested up to the maximum plastic strain of >6%. What was the actual maximum compression strain? 6% compression strain is too low to claim that the alloys are ductile. It would be great if the authors report at which compression strain (in the range of at least 0-30%) the alloys fail and what was the type of fracture. Generally alloys, which fracture strain during compression deformation is less than 10%, show brittle type of fracture (cleavage or intergranular fracture) and they are brittle during tensile deformation.

6) Ductility analysis based on Considere criterion (lines 176-188) can only be applied to ductile materials, which form necking during tensile testing. The authors must prove first that the alloys show ductile type of fracture. For this, please extend deformation until the samples fail and then report compression fracture strain, how the samples fail (explosive, shear or cleaved type) and also provide images of the fracture surfaces. The authors' brief mentioning that micro-cracks were developed parallel to the compression direction (i.e. perpendicular to the tensile stresses) indicates that the alloys are brittle. The authors statement on "exceptional compressive ductility" must be proven by deforming the samples to much larger strains than 6%.

The referee is correct with the statements on metallic-intermetallic materials and the different behavior in compression and tension. We are aware of these complications and, *thus, adopted the manuscript to highlight the restriction of a straight transfer of the present results to tensile behavior; specifically considering the longitudinal cracks obtained in our investigations. Particularly, we highlight the role of grain boundaries in our conclusion section – it for sure a subject that needs to be investigated in future. However, we do not consider the compression test setup a relevant setup to probe and assess the role of grain boundaries properly.*

We deliberately stopped the compression tests in our investigations when work hardening of the specimen became solely determined by the increase of the cross-sectional area of the specimen, not by the intrinsic behavior. *The data beyond the strains we stopped is indicative of the specimen behavior only, not the materials behavior in first place.*

As we fully agree on the role grain boundaries for extended plasticity in tension, the microstructural conditions need to be tailored with respect to grain size, shape and distribution to reveal the full potential. *The metallurgical options and challenges associated to this are part of the revised manuscript to outline that this is not within the scope and possibilities of the present investigation.*

The amount of work hardening and the activation of deformation twinning is unique and not obtained in any Strukturbericht A2 alloy so far. The observations on the additional, homogenized condition of Cr-36.1Mo-3Si highlight the huge optimization potential with regard to strength, work hardening and most probably tension/compression ductility by tailoring the interplay between twinning and dislocation slip. *We added relevant statements.*

Though, the microstructural tailoring is not straight forward and currently out of the scope of the research for this study. The inhomogeneous microstructural condition of the Cr-Mo-Si alloy presents the most fine grained condition that is currently available – still with a grain size > 100 μm . As shown in the revised version of the manuscript, the homogenization treatment results in a substantial grain growth. To conclude on relevant tension behavior, on relevant fracture mechanisms or to allow for the direct transfer to tensile behavior, fine-grained material is required.

This might follow classical cast metallurgical routes with subsequent thermomechanical treatment. This processing route may benefit from interstitial contents similarly low as in our study if proper crucible-mold combinations are identified. The alternative would be a powder metallurgical synthesis. The major restriction will be the limitation of interstitial contamination levels.

Both are the opportunities provided by the findings in our manuscript. *We added the relevant conclusions on these materials engineering aspects to the manuscript to encourage the community to contribute to these research activities as this development is out of the possibilities of the present article.*

Other, less critical comments.

7) Line 11 “at low temperatures.” Could you be more specific? Do you mean room temperature?

We clarified the statement.

8) Lines 108-120: “The specific mass loss of the samples is between -5.5 and -6 mg/cm².” If you said mass loss, then do not use the negative sign in the respective value. The mass loss of -5.5 actually indicates the mass gain of 5.5 .

We corrected the statement.

9) Lines 189-205 and below. Growth of the Cr₂O₃ scale should result in the mass gain, while the mass loss is observed. Based on the thickness of the scale, one can estimate how much Mo was lost during oxidation in both alloys.

We gratefully acknowledge this remark. *We confirmed the consistency (within accuracy and applicability model assumption) of the data by transferring mass change to scale thickness and vice versa. We added the relevant data to the manuscript.*

The statement on line 196 “no protective scale has formed” seems to contradict to the experiment. A continuous scale of the same Cr₂O₃ forms in both alloy. Should discuss why the scale thickness increases linearly in the binary alloy. Why Cr₂O₃ continuous scale does not slow down oxidation in this alloy. How the MoO₃ evaporates through continuous Cr₂O₃? Why much thinner layer of the same Cr₂O₃ formed in the Si-containing alloy is a more protective scale? Why Mo is less volatile in the Si-containing alloy? All these questions should be discussed/answered.

For the Si-free alloy, the Cr₂O₃ scale is by far not continuous and a layered mockup is observed. This is addressed in the manuscript. With respect to the protectiveness, we rely on the growth law of this layered Cr₂O₃ which indicates an *unlimited, linear* growth and the disparity between experimental mass change (negative) and assessment of the mass gain expected by the oxide growth.

We modified the manuscript to highlight that the layered mockup of the scale on the Si-free alloys may result from the repetitive (i) consumption of Cr by Cr₂O₃ formation and according subsurface enrichment in Mo and (ii) oxidation of enriched Mo to MoO₃.

10) Line 217 “cubic growth law” but line 259 “The scale growth occurs almost parabolic.” These statements should be corrected. The authors should provide the oxidation equation used for the analysis: $dn = kt$. Then, in the second instance, the corrected phrase is “The scale growth follows the quadratic law”.

Over the course of the incorporation of the new results on the homogenized alloy, we have adopted the fitting procedures considering the experimental uncertainties more clearly. The current assessment is thus more balanced and provides relevant insights into apparent growth laws and the limits of the mechanistic interpretation.

11) Lines 231-232: “Mo addition to Cr-Si alloys promotes the formation of SiO₂ consistent with results from the present study.” I am not sure if this statement is correct, because SiO₂ was not detected experimentally in the studied alloy at 800°C.

This was tackled by the additional investigations and the statements are updated.

12) Paragraph lines 224-238. The authors did not answer the main question: why MoO₃ does not form or forms in a very small amount in the Si-containing alloy, while this process dominates in the alloy without Si.

This is addressed in the novel version of the manuscript.

13) Lines 284-295. What was the texture of un-deformed samples parallel to the compression axis. I doubt if any noticeable texture would develop after only 6% compression strain.

The samples were randomly cut from the button. Thus, no direction of reference is available for the as-cast condition (no unique solidification direction is seen for example in the updated micrographs of the initial conditions in Fig. 1). A direct comparison of the texture prior to the compression test and after was not possible in the present study. As highlighted in the text, the assessment is currently limited as no quantitative conclusions can be drawn and the qualitative conclusions need to be verified.

The potentials and challenges of developing fine-grained versions of the present alloys to test these trends quantitatively are stressed in the revised version of the manuscript.

14) Line 307: “some short microcracks, located predominantly parallel to the compression direction,” Crack developed parallel to compression direction (i.e. perpendicular to the tension stresses) after 6% compression strain is an indication of brittleness of the alloy samples.

The potentials and challenges of developing fine-grained versions of the present alloys to quantitatively test these trends are stressed in the revised version of the manuscript.

15) Line 312: “Fundamentally, deformation twinning competes with dislocation slip” These are not competing but complimentary processes.

We agree and modified the occurrences.

16) Lines 359-366. It is clear that the authors missed the main point: the presence of Si slows down formation of volatile MoO₃. The reason for this should be uncovered and discussed.

This was tackled by the additional investigations and the statements are updated.

17) Materials and Methods section. What was the mass of arc melted samples? Why homogenization treatment was not used to get rid of the dendritic structure?

This was addressed by additional experiments and the missing information was added.

Referee #2 (Remarks to the Author):

In the paper on “Overcoming barriers for high temperature alloys: oxidation and ductility” the authors present a systematic study of the behavior of Cr-Mo alloys with and without the addition of a minor level of Si. The results of the study would be of interest to the community examining high temperature refractory metal alloys. Before the paper can be considered further there are several issues that require clarification and revision.

First, as the authors point out in the introduction if their alloy is intended to enable an increase in operating temperature beyond the capability of current Ni base superalloys, why did they only conduct the testing up to 1100°C? At 1100°C the Ni base superalloys already perform well. In order to displace the Ni base superalloys, the evaluation must be done at a higher temperature of 1300 or 1400°C.

We agree with the reviewer to some extent. With respect to the actual performance of Ni-based superalloys under mechanical load, the maximum capability for *single-crystalline alloys* is often denoted as 1050 to 1150°C (most reliable data for CMSX-4 is for 1050°C) substrate temperature, thus below thermal and environmental barrier coatings and when active cooling of the part is applied. Even small increase by several 10 K to 1100°C substrate temperature (still employing the current engineering option to keep combustion temperature way higher) would already display a considerable step in terms of efficiency. A further increase should be the long-term development goal of course. *This may be likely as all metallurgical optimizations are feasible starting from a proper single-phase system, like the one presented in this investigation.*

Indeed, the research over minimum the last two or even three decades provided alloy candidates capable of providing resistance up to 1150 or even up to 1400°C. However, only with respect to *either* resistance against oxidation or creep resistance. A combination of both in the entire temperature range is still scarce; specifically in conjunction with deformability at room temperature, *no solution exists not even in compression. Please note that for the present material candidate, active deformation mechanisms are identified at room temperature that are obtained in metallic materials at low temperatures. This is not the case for any high intermetallic or ceramic material.*

This is the scope of our contribution. The manuscript provides a base alloy (system) capable of withstanding elevated temperatures without the need of oxidation protection by suitable coatings at 1100°C for extended periods of time. It exhibits compression deformability at room temperature *with a yet unmet deformation mechanism combination and work hardening capability.*

For sure, there are challenges with respect to the microstructural tailoring of the microstructure to benefit from these properties in tension and to obtain relevant part dimensions, please see also our response the first referee on the role of grain boundaries.

With respect to creep performance, the current condition of the alloys won't meet competitive requirements as a *single-phase, polycrystalline Strukturbericht A2 crystal structure* is established. However, the relevant basis for further alloy development and further strengthening is set in distinct contrast to other investigations where no base oxidation resistance or ductility was provided; two properties that rarely get better when alloy or microstructure modifications are applied.

We modified the manuscript to highlight the aforementioned consideration more strongly. The relevant conclusions for the community are stressed.

This is an important point. For example, while the chromia scale appears to work in the Cr-36.1Mo-3Si alloy at 1100°C, it is well known that chromia will lose integrity and protection above 1100°C due to the development of volatile CrO₃. In fact, this issue is mentioned on line 281 for 1100°C. Thus, the challenge of overcoming oxidation at high temperature will not be met by the Cr-36.1Mo-3Si alloy.

Indeed, future development will for sure need to tailor the appearance of the chromia scale – potentially by establishing Al modified chromia or even alumina with substantial amounts of Cr. This development trend is possible with the current alloy system. *We added the remark to the conclusions of the alloy assessment.*

In figure 1 the as-cast samples show a clear dendritic microstructure. Since there is no indication of a homogenization treatment, it appears that the oxidation testing was done on as-cast samples. However, during the long-time oxidation tests at high temperature the as-cast microstructure can change and the change can confound the analysis of the oxidation behavior. This change can also influence the apparent parabolic and cubic kinetics shown in figure S4 which by the way is not explained in the manuscript. For example, how do the rate constants and exponents relate to the proposed oxidation mechanism in figure 3?

We manufactured additional material in homogenized condition and analyzed these new microstructural conditions in the same way as described in the initial manuscript.

All relevant key conclusions were confirmed. The additional experiments were added to the manuscript in a consistent way. Specifically with respect to the growth laws and mass changes, a more balanced description considering uncertainties of the determined quantities and the fits were established.

It is not clear if the mechanical tests were conducted in air, argon or vacuum. In figure 4 a and b there appear to be precipitates (silicides?) at some of the grain boundaries which implies according to figure 3d that the sample oxidized.

The tests were conducted in air by induction heating. This is outlined in the experimental description. The test periods were short enough and the protectiveness of the alloys sufficient to prevent noticeable oxidation of the samples. No formation of secondary phases within the compression test specimens were observed.

Moreover, while the development of grain boundary cracking may not lead to complete failure under compression, the cracks would reduce the ductility under tension and also result in poor fatigue performance. This behavior does not support the claim of overcoming the ductility challenge. Unless these points are fully addressed the authors' conclusions can not be accepted.

The referee is correct with the statements on the role of grain boundaries tested in compression and tension. We are aware of these aspects and *adopted the manuscript to highlight the restriction of a straight transfer of the present results to tensile behavior; specifically considering the longitudinal cracks obtained in our investigations.*

However, *the amount of work hardening and the activation of deformation twinning is unique and not obtained in any Strukturbericht A2 alloy to this amount.* It allows unique opportunities to tailor the mechanical response specifically by modification of grain size.

The observations on the additional, homogenized condition of Cr-36.1Mo-3Si highlights the huge optimization potential with regard to strength, work hardening and most probably tension/compression ductility. *We added relevant statements.*

Alexander Kauffmann

Nature
Referees

Institute for Applied Materials (IAM-WK)

Head of Business Unit:
Prof. Dr.-Ing. Martin Heilmaier
Prof. Dr. rer. nat. Astrid Pundt
Prof. Dr.-Ing. habil. Volker Schulze

Engelbert-Arnold-Straße 4
76131 Karlsruhe, Germany

Phone: +49 721 608 46594
Email: alexander.kauffmann@rub.de
Web: www.iam.kit.edu/wk

Official in charge: Alexander Kauffmann
Date: 05/09/2025

Revision of an article

Dear referees,

please find in this electronic re-submission our revised manuscript on “Overcoming barriers for high temperature alloys: oxidation and ductility” (F. Hinrichs et al.).

We greatly appreciate the thorough review by the referees. We conducted the additional experiments as suggested by the reviewers. The additional straining to large strains in compression proves the ductility statement on the as-cast condition while the homogenized material exhibits lower plastic strains. This stresses the optimization potential via microstructure design of the present alloy as proposed in the initial version of the article. With respect to the underlying oxidation mechanism, we incorporated a novel evaluation scheme on the ratio of oxidized Cr (passivating) and Mo (deteriorating) ions. As the evaluation might be sensitive to the evaluation of mass changes and the scale thicknesses incl. the respective experimental or statistical uncertainties, we performed additional experiments and measurements of these quantities. The results prove the improved selectivity of beneficial Cr oxidation over the detrimental oxidation of Mo as suggested by the Ellingham diagram which is now included as well.

We are convinced that the key results of our study are verified to an extent that allows the community to proceed with the further engineering development of this novel type of alloy. Furthermore, we appended mechanistic interpretations of the finding in the current stage of research. We have added a forecast to the research and developments that are required to take advantage of the observations from our investigations.

We agree with the reviewers on the required development that is needed into such a groundbreaking observation. Therefore, we strengthened and checked the manuscript for a balanced assessment of opportunities and challenges with further development. However, specifically by proving the ductility in tension a much greater

effort is needed (by the community as we alone might not be able to capture all possible synthesis and processing scenarios) than what is feasible for the present study (and our group now). The results already indicate that revealing the full potential requires deliberate microstructural tailoring which is beyond our intention to provide the observation and its detailed characterization of it to the community. We appended the options for synthesis and processing clearly in the manuscript.

Please find the point-by-point description of our revisions and response in the following. The according changes in the manuscript are marked up by red color.

Referee #1 (Remarks to the Author):

The authors correctly addressed some of my comments in revised manuscript and conducted additional experiments, which is much appreciated. However, the authors responses to my comments 5) and 6) are not satisfactory and I cannot accept them.

In my comment 5) I pointed out that 6% compression deformation is too low to claim that the alloy is ductile and I suggested the authors to compression deform the alloy sample to fracture and report the fracture strain. This should be a simple experiment, which would provide an important information on the compression ductility and brittleness of the alloy. This information is critical, as the authors claim that the studied Cr-36Mo-3Si alloy is both oxidation resistant and ductile. The second statement should be proven experimentally. Currently, the authors failed to prove this and, therefore, they main authors "achievement," synergy of good oxidation resistance and good ductility, is still questionable.

From the first revision stage:

5) Mechanical properties. Why compression deformation stopped after 6% strain? Later the authors state that the samples were tested up to the maximum plastic strain of >6%. What was the actual maximum compression strain? 6% compression strain is too low to claim that the alloys are ductile. It would be great if the authors report at which compression strain (in the range of at least 0-30%) the alloys fail and what was the type of fracture. Generally alloys, which fracture strain during compression deformation is less than 10%, show brittle type of fracture (cleavage or intergranular fracture) and they are brittle during tensile deformation.

We have conducted the requested experiments:

- Fine grained, as-cast (AC) material exhibits plastic strains of 9–15% at maximum stress in compression. Total plastic strain at interruption is beyond 15%.
- Coarse grained, homogenized material (H) exhibits 4–6% at maximum stress. Total plastic strain at interruption is minimum 9% for this condition.
- The material **did not disintegrate/shatter** in either of the microstructural conditions until the interruption of the tests. Therefore, no fracture surfaces could be analyzed.

Ductility in the sense of the request is confirmed. This information was added to the manuscript, l. **191-201 as well as Supplementary Material Fig. S2.**

This behavior is consistent with the trends outlined in the original draft:

- Dislocation slip requires comparably high stress values due to strong solid solutions strengthening (note the 9% lattice parameter difference between Cr and Mo). First estimates of the onset stress in

Cr-36.1Mo-3Si is between 1500 and 1900 MPa (depending on screw or edge dislocation control). The theoretical assessment of solid solution strengthening is complex as edge and screw dislocation contributions need to be considered in contrast to dilute bcc alloys where only screw contributions are relevant.

- Deformation twinning is already active at stresses between 400 to 600 MPa. This is surprisingly low for bcc metals and alloys. It is furthermore confirmed to be grain size sensitive in the sense that fine grained microstructural conditions lead to suppression of twinning (higher onset stresses for twinning).

The current trends in ductility suggest an optimal intermediate grain size to obtain:

- Initiation of sufficient twinning to foster work hardening by dislocation interaction → not too fine grained
- Sufficient grain refinement to obtain internal stresses to activate dislocation slip and avoid crack initiation at tips of twin lamellae → not too coarse grained

This assessment and the conclusions are included in the manuscript in **I. 546-561**.

My comment 6) has not been addressed either. Considere criterion can only be applied to tensile testing of ductile materials and it estimates strain at which plastic instability (necking) may start to form during tensile test. It cannot be applied to the compression tests of a brittle alloy. As I mentioned in my comment, the authors must prove that the alloy shows ductile behavior and ductile type fracture. But they failed to do these simple additional experiments. This additional information is needed for this manuscript to prove the authors statements about "overcoming barriers for high temperature alloys: oxidation and ductility". Otherwise, the paper title is misleading.

From the first revision stage:

6) Ductility analysis based on Considere criterion (lines 176-188) can only be applied to ductile materials, which form necking during tensile testing. The authors must prove first that the alloys show ductile type of fracture. For this, please extend deformation until the samples fail and then report compression fracture strain, how the samples fail (explosive, shear or cleaved type) and also provide images of the fracture surfaces. The authors' brief mentioning that micro-cracks were developed parallel to the compression direction (i.e. perpendicular to the tensile stresses) indicates that the alloys are brittle. The authors statement on "exceptional compressive ductility" must be proven by deforming the samples to much larger strains than 6%.

In the revised version of the manuscript, we clarified the role of Considère criterion for tensile tests to be relevant only. True work hardening in compression reflects intrinsic work hardening of the material and thus the potential against localization in tension. That ductility in tension *can* be limited by other factors is stressed in the manuscript. We checked the mentioning of this restriction, for example **I. 227-246, I. 448-450, I. 552-555**.

It is relevant to note that the tests requested by the reviewer (9–15% plastic strain at maximum stress for AC vs. 4–6% for H) and the trends in (true) work hardening (substantially higher for AC and smaller for H) are consistent. The results suggest an optimal microstructural condition of intermediate grain size to be established. We stressed this in the recent version of the manuscript.

I am also not satisfied with the authors responses to my comment 2). From one side, they included the non-equilibrium Scheil assessment, which is much appreciated. On the other side, discrepancy of the Scheil model

with experimental results for Cr-Mo-Si they tried to explain by the absence of the thermodynamic data for the Sigma-(Cr,Mo,Si) phase in the used database. However, if the studied Cr-Mo-Si alloy is truly single-phase BCC structure, as the authors state, the absence of some intermetallic phases in the database should not affect the CALPHAD calculations towards the BCC phase.

We have removed the ambiguous statement. The intention of the part was to highlight the current inconsistency of the thermodynamic assessments to lay out the paths for the future evaluation of the current and other alloys within the Cr-Mo-Si system. We have prepared a clear overview of relevant phase data in Fig. S1a of the Supplementary Material, a direct comparison to the current calculation from Pandat in Fig. S1b incl. a suggestion of how the σ -(Cr,Mo,Si) might change the liquidus surface and, thus, the involved invariant reaction during solidification of similar alloys. This is discussed in I. 112-123.

Lines 11-13. The authors continue to state that the Cr-36Mo-3Si shows “sufficient ductility at low temperatures” and thus shows “outstanding and yet unmet property.” First, please be specific and replace “low” with “room”.

We removed the ambiguous statements from the entire manuscript.

Second, 6% compression strain, which also includes elastic strain, is not sufficient to state that the alloy shows “sufficient ductility” at room temperature. Will the alloy show yielding during tensile testing or it will fracture during elastic loading? I bet the latter will happen. At what compression strain it fractures? What are the fracture modes for this alloy? I asked similar questions in my first review. The authors provide no answer and no additional experiments to disprove that the alloy is actually brittle.

Lines 51-54: I would agree that the oxidation behavior of the Cr-Mo-Si is outstanding; however, I disagree with the statement that the alloy shows “sufficient ductility at low temperatures.” The authors did not prove that the alloy shows “outstanding and yet unmet property combination” (oxidation and ductility).

Please see the novel results from the requested additional tests as described above.

Lines 95-103. This is incorrect interpretation of the discrepancy between CALPHAD calculations and the experiment. If BCC is an equilibrium phase, then absence of any other phases from the database should not affect the result: Gibbs free energy of the BCC phase should not increase relative to the Gibbs free energy of the other phases, present in the database due to the absence of some of the other phases!

We have removed the ambiguous statement and appended an overview about the effect of the σ -(Cr,Mo,Si) that needs to be considered in future thermodynamic assessments of the Cr-Mo-Si system to correctly map the Cr-Mo-Si system, see the above statement on this aspect.

With respect to multiphase equilibria: These are determined by the common tangent planes to Gibbs free energy curves of the participating phases. These common tangent planes are altered by choice of phases (and thus Gibbs free energy curves) considered. In the present case, the current thermodynamic assessment predicts a two-phase eutectic trough that completes solidification under Scheil condition for Cr-36.1Mo-3Si. However, in the region of this predicted eutectic trough, the existence of the congruently melting σ -(Cr,Mo,Si)

phase is experimentally confirmed but not yet considered in the current thermodynamic assessments. Consequently, the type of invariant reaction between $(\text{Cr,Mo,Si})_{\text{ss}}$ and $\sigma\text{-(Cr,Mo,Si)}$ in the compositional range of the present alloys needs to be re-assessed in future work. It is likely *not* a eutectic reaction as no indication of a kinetically fast solidification reaction is found in our investigations. As this does not belong to the core scope of our study, we decided to leave out this elaboration from the draft.

Table 1. Please provide the explanation why the amount of O and N decreased after annealing in Ar at 1600C/48 h.

Due to the initial requests by the reviewers, additional material was produced; this information is included in the experimental section. The O/N content of Cr based material is determined by the O/N content of the pure Cr and O/N uptake from the Ar during the homogenization. We optimized the synthesis by improved raw batches and better wrapping in gettering materials. We added this information to the manuscript, **I. 574-585**.

Lines 135, 137 (related to my comment 8) in the first review). If you write "loss" then the value must be positive. Otherwise, the loss of -6 mg/cm² indicates +6mg/cm² mass increase!

We have removed all instances of "loss" in conjunction with negative values of mass change.

Sentence in Lines 156-157 should be revised. If spallation and evaporation events are absent, then what is responsible for the "small mass change" of this alloy at 800C? How was the mass change measured? Was the mass of spalled material included or it was measured separately, using the mass of the remaining sample? This should be described in details in Methods.

We have clarified the statement on the measurement from "The samples were oxidized in Al₂O₃ crucibles and weighed with the crucibles between oxidation intervals." to "The samples were oxidized in Al₂O₃ crucibles and weighed within the crucibles between oxidation intervals.", **I. 595**. We appreciate the remark by the reviewer and removed the spallation statement from the assessment in **I. 164-166**.

Lines 186-191. The authors ignore my comments 5) and 6) in my previous report regarding the validity of the compression tests with low strain levels (not exceeding 6%) and using Considere criterion. I still insist that compression strain should be extended to the fracture event to confirm that the material is indeed ductile. Or please conduct tensile tests and show that the alloys will not fracture during elastic loading.

We have conducted the requested experiments. Please see the detailed answers above.

Line 195: "For all test temperatures up to 800C ..." What is this? The samples were tested at two temperatures, RT and 900C only!

As indicated in Fig. 2c and described in the experimental, tests were conducted at RT, 100, 200, 400, 600, 700, 800, 900 °C. Additional tests at 1100 °C were conducted upon the first reviewer requests (on another machine due to the capability limits of the induction setup). For the sake of clarity, RT and 900 °C stress strain curves were exemplary selected for Fig. 2b. We have improved the caption of Fig. 2b and the experimental, **I. 616-621**.

Lines 208-212: What is the reason of twice lower strength at room temperature after homogenization treatment? This noticeable drop in strength cannot be explained by grain growth.

The reviewer is correct that grain boundary strengthening cannot account for this difference. The Hall-Petch constant might be similar to $800 \text{ MPa}\sqrt{\mu\text{m}}$ [Z.C. Cordero, B.E. Knight, C.A. Schuh, International Materials Reviews 61 (2016) 495-512] for pure Cr. The strength contribution changes from 80 MPa for about 100 μm in the as-cast (AC) condition to 20 MPa in the coarse-grained, homogenized condition (H, > 1500 μm). The experimentally obtained difference is larger, even when the plateau region between 400 and 700 °C is considered, i.e. 230 MPa. We have added this information to the manuscript, I. 217-220. We appreciate this remark.

The two microstructural conditions of Cr-36.1Mo-3Si exhibit distinctly different deformation behavior. The 1% offset yield strength of the AC condition resembles the typical temperature strength dependence of bcc metals and alloys when dislocation slip is mediated by thermally activated kink pair formation and propagation: (1) substantial strength decrease from room temperature to 400 °C, (2) plateau strength at intermediate temperatures between 400 to 700 °C and (3) marked drop by the onset creep deformation at high temperatures beyond 700 °C. Thus, dislocation slip considerably contributes to strength and plasticity. In contrast, deformation twinning is considered an athermal deformation mechanism which is consistent to the temperature dependence of the homogenized condition H, see I. 466-474.

The proportion of deformation twinning to (lower) strength and plasticity in the H condition is larger. This is reasonable based on the following considerations as described in the manuscript:

- Strong solid solution strengthening of dislocation slip in Cr-Mo due to large lattice parameter variation, see I. 490-503
- Grain refinement leads to strengthening when dislocation slip is active
- Deformation twinning is active at surprisingly low stresses between 400 to 600 MPa, see I. 475-489

We opted for assessing $\sigma_{1\%}$ as plastic deformation occurs serrated. Fig. 2b indicates that the strain hardening behavior is different for the two microstructural conditions. This amplifies the difference between the two conditions when $\sigma_{1\%}$ is considered instead of $\sigma_{0.2\%}$. The actual onsets of plastic deformation are much closer to each other than indicated by $\sigma_{1\%}$ (determined by deformation twinning), though difficult to determine.

Lines 219-221. This conclusion should be proven using tensile tests. As mentioned above, Considere criterion cannot be applied to compression tests.

Please see the detailed responses on this issue above.

Figure 2. Please plot Figure 2(b) in coordinates true stress versus true strain.

We have changed the diagram to true stress-strain curves.

Lines 232-234: "the scale is identified as monolithic Cr₂O₃" but "The oxide scale is composed up of porous, layered microstructure of Cr₂O₃." These statements contradict each other. Should be revised. The word "monolithic" is used in several instances in the manuscript.

We have removed monolithic.

Lines 360-373. Discussion on the fiber texture is misleading. Answering to my comment 13), the authors said that the samples were extracted from the arc-melted button in random orientations and their texture was not analyzed. Therefore, the texture observed in the compressed sample can be an original texture, not formed after deformation. I would recommend removing all the "texture" data from the manuscript in this case.

We have removed the term texture and highlighted "slightly preferred orientations" in **I. 423**. The lack of statistics is clearly indicated by "Even when considering the low number of grains probed in the present samples" as well as "However, the statistical relevance needs to be confirmed with fine grained material.", **I. 429-430**.

Lines 463-473. Oxidation mechanism and role of Si. At 3 at.% Si, could the authors estimate how much oxygen Si can get and prove that this amount of SiO₂ formed can explain the observed reduction of the Cr₂O₃ growth rate and no formation of MoO₃? The current vague explanation of the oxidation process in Cr-36Mo-3Si, as well as insufficient mechanical testing experiments, does not deserve the publication in Nature.

We appreciate the remark by the reviewer. We have conducted additional experiments to further prove effect by Si. We have included the relevant reaction in the Ellingham diagram in Figure 6 which indicated that the oxidation of Cr to Cr₂O₃ and Mo to MoO₂ are close to each other. By adding 3 at.% Si, the selectivity of the reaction seems to be shifted towards the oxidation of Cr allowing for passivation.

By evaluating statistically relevant mass changes (multiple samples under conditions without spallation, namely both alloys tested at 800°C up to 100 h) as well as scale thicknesses from multiple sides of tested specimens, we were able to recalculate on the Cr and Mo ions consumed in oxidation. The assessment proves the selectivity change between the reactions by Si. Furthermore, the assessment suggests a critical Cr to Mo ion ratio of approximately ≥ 3 to achieve passivation. This can be achieved by (i) low enough oxidation temperature, (ii) low enough oxygen partial pressure, (iii) low enough Mo content, and (iv) high enough Si content. This needs to be confirmed in future work. The assessment is added to **I. 384-406 and 601-610**.

Referee #2 (Remarks to the Author):

In the revised manuscript the authors have addressed all of my review comments. The additional experiments on homogenized samples including the APT measurements have clarified the oxidation behavior. In addition, a note has been added concerning the need to address the performance above 1100°C. This will be a challenge and must include resistance to water vapor attack. The authors have also added a clarification concerning the difficulty in relating compression test behavior to tensile behavior which is another challenge. I appreciate their opinion regarding the replacement of Ni base superalloys by their alloy, but I maintain my opinion that the current performance of their alloy does not qualify as a replacement. Much more development is required for their alloy.

We appreciate the assessment of the reviewer. We agree with the reviewer that further development is required to exploit the full potential of the present alloy. The relevant aspects and the outlook are appended in **I. 546-561**. The tremendous development of Ni-based superalloys and turbo engine technology in general to the status, however, also traces back to incredible research investment over more than 85 years, starting at much lower capability than available today. This might put the report of our observation and its detailed characterization into perspective.

[REDACTION]

Taken from R.C. Reed, The Superalloys: Fundamentals and Applications, Cambridge University Press, Cambridge, 2006. <https://doi.org/10.1017/CBO9780511541285>.

Alexander Kauffmann